# Unravelling single-cell DNA replication timing dynamics using machine learning reveals heterogeneity in cancer progression

**Joseph M. Josephides** ⓘ **& Chun-Long Chen** ⓘ ✉

Genomic heterogeneity has largely been overlooked in single-cell replication timing (scRT) studies. Here, we develop MnM, an efficient machine learning-based tool that allows disentangling scRT profiles from heterogenous samples. We use single-cell copy number data to accurately perform missing value imputation, identify cell replication states, and detect genomic heterogeneity. This allows us to separate somatic copy number alterations from copy number changes resulting from DNA replication. Our methodology brings critical insights into chromosomal aberrations and highlights the ubiquitous aneuploidy process during tumorigenesis. The copy number and scRT profiles obtained by analysing >119,000 high-quality human single cells from different cell lines, patient tumours and patient-derived xenograft samples leads to a multi-sample heterogeneity-resolved scRT atlas. This atlas is an important resource for cancer research and demonstrates that scRT profiles can be used to study replication timing heterogeneity in cancer. Our findings also highlight the importance of studying cancer tissue samples to comprehensively grasp the complexities of DNA replication because cell lines, although convenient, lack dynamic environmental factors. These results facilitate future research at the interface of genomic instability and replication stress during cancer progression.

DNA replication is a fundamental biological process in which a cell creates an identical copy of its genome to ensure the accurate transmission of its genetic information to daughter cells during the synthesis (S) phase of the cell cycle. Despite the robustness and tight regulation of this process, errors can occur leading to inexact, over- or under-replication. Therefore, genomic alterations, including DNA copy number variations (CNVs)[1–4], point mutations[5–7], and other structural variants, are also common in normal cells[8–13]. CNVs refer to abnormal fluctuations in the number of copies of specific genomic regions and can be caused by various cellular processes, such as DNA replication errors, lack of replication factors, chromosomal instability, and DNA damage[14]. Dysregulation of DNA replication is associated with various human diseases, including cancer[15,16]. Replication stress, which occurs when cells encounter obstacles that prevent the successful execution of the replication programme, can cause genomic instability and promote tumorigenesis, making it a hallmark of cancer[14,17]. The study of CNVs and DNA replication alterations is particularly important in cancer because they can affect gene expression and function, potentially leading to tumour initiation, progression and therapy resistance. They also serve as biomarkers for cancer diagnosis, prognosis, and treatment response[18,19]. However, methods specifically adapted to study DNA replication in cancer cell populations are lacking due to the high intra-tumoral heterogeneity.

CNV detection from next generation sequencing (NGS) data has traditionally involved measuring the number of DNA copies per genomic region using bulk sequencing data. This has been and remains

Institut Curie, PSL Research University, CNRS UMR3244, Dynamics of Genetic Information, Sorbonne Université, Paris, France.
✉e-mail: chunlong.chen@curie.fr

a popular approach to identifying CNVs in cancer samples. However, in bulk sequencing, the genetic material of different cells is mixed and sequenced, thus providing only an average image of the genomic alterations across all tumour cells. This approach may not allow the identification of subclones with distinct genomic profiles that could be critical for understanding CNV origin and evolution in cancer. Although efforts have been made to develop innovative mathematical and computational approaches to estimate tumour purity and leverage variant allele frequency in bulk sequencing data, the accurate detection of intra-tumoral genomic heterogeneity[20,21], an important aspect of evolving tumour populations[11,22], is still challenging because bulk data do not provide single-cell resolution. Therefore, this classical approach might not capture the full degree of genomic heterogeneity in tumours, limiting the identification of genomic changes that are necessary to understand and study cancer progression, drug resistance and therapeutic bottlenecks.

Recently, the advent of single-cell technologies has revolutionised cancer research. Particularly, single-cell genomics has enabled the study of cancer heterogeneity and the identification of rare cell populations that may be responsible for tumour initiation and therapy resistance. This technology has been used to characterise CNVs and other genomic alterations in individual cells, revealing the extent of intra-tumoral heterogeneity and clonal evolution in cancer[22–30]. This has led to the discovery of genomic subpopulations (i.e. groups of cells with distinct CNV signatures compared with other cells from the same sample). Yet, intra-tumoral heterogeneity is still overlooked when studying DNA replication.

Replication Timing (RT) is a key metric in DNA replication studies. RT refers to the order in which different genomic regions are copied during the S phase of the cell cycle. RT is highly regulated and correlates with other cellular processes, including gene expression[31], DNA methylation[32], chromatin structure[33] as well as the 3D organisation of chromosomes[34]. Single-cell whole genome sequencing (scWGS) has allowed upgrading RT studies and the emergence of more detailed analyses of mammalian replication dynamics at the genome-wide level[32,34–38]. Until recently, single-cell RT (scRT) studies were limited by the number of cells that could be analysed in each sample due to technical constraints. Some recent studies that have overcome these hurdles[32,37,38] confirmed that cell-to-cell heterogeneity can be observed in a single sample. We previously showed that it is possible to distinguish subpopulations based on their CNVs and extract distinct replication patterns using scWGS data from a single heterogenous cell line sample[37]. However, this process was not automated and consequently, RT profiles are not routinely disentangled in tumours.

Here, to democratise RT studies in complex cancers, we develop a machine learning-based tool that we called Mix 'n' Match (MnM). This automated tool exploits the CNVs of single cells in mixed cell populations to cluster cells based on similarity, thus reconstructing subpopulations and discovering replication states in silico. By grouping cells into subgroups based on their cell cycle phase (replicating *versus* non-replicating) and genomic composition, we observe distinct RT trajectories for different subpopulations from cell lines and tumour samples. We rigorously train a ready-to-use supervised machine learning model to identify replicating cells and apply unsupervised learning to identify cell subpopulations. By analysing 119,991 human single cells, we provide a large source of heterogeneity-resolved scRT profiles, which unravels replication timing dynamics and reveals an additional layer of heterogeneity in cancer progression.

## Results

### Accurate copy number imputation of unsynchronised single cells

Due to technical limitations, scWGS frequently has a lower read coverage across the genome (i.e. <1X, Supplementary Tables S1-3) compared to bulk sequencing, resulting in regions of copy number data missing randomly across the genome. To address this issue, we used the k-Nearest Neighbors (KNN) imputation technique, a data completion method that considers the closest cells in terms of genome-wide copy number profiles[39]. To take into account rare CNVs and replication events, we used a weighted copy number distance for the KNN imputation that generated an imputed value proportional to the closeness of the genome-wide copy number profiles between cells based on Euclidean distances (Fig. 1a, b). For each missing value, we used the existing copy numbers of the closest five single-cell profiles for the same region to fill in the missing data (see Methods for details).

We proceeded to empirically validate this method by simulating sparse single-cell copy number matrices. We introduced random voids within the 100 kb window single-cell matrix of a mixture of replicating and non-replicating MCF-7 cells ($n = 2321$ cells; $n = 1288$ genomic regions), a breast cancer cell-line with a large number of CNVs (mean ploidy: 3.73)[37], after removing all regions that already contained missing values. We randomly removed between 5% to 55% of copy number values through increments of 5%. The KNN imputation method predicted and integrated these missing values with an average accuracy of 83.96%, thereby reconstructing the single-cell copy number landscape. Our findings showed an invariance rate, defined as the total percentage of intact values in the whole matrix, ranging from 99.21% to 90.92% for 5% to 55% of missing values, respectively (Supplementary Fig. S1a). These values were significantly higher (paired *t*-test, $p = 7.67\text{e-}24$ and $p = 5.34\text{e-}25$, respectively) than the median (ranging between 97.44% and 71.80%) and the random invariance rates (ranging between 96.12% and 57.25%), underscoring the robustness of the KNN approach. Furthermore, the imputed values with an absolute difference of ≤1 copy number, compared with the original values, ranged between 99.92% and 99.01%, indicating that the vast majority of the errors introduced through this process were not radically inaccurate, even when more than half of the dataset contained missing values. We repeated the evaluation of the KNN imputation method using scWGS data of HeLa S3 cervical cancer cells (mean ploidy: 2.87, $n = 459$ cells) and JEFF B cells (mean ploidy: 1.94, $n = 952$ cells) (Supplementary Fig. S1b, c) and obtained similar invariance rates (99.28–91.33% and 99.61–95.43%, respectively). These random simulation results confirmed that our KNN imputation method can robustly produce accurate results using scWGS data obtained from normal and cancer cell types with various ploidy levels.

To extract as much information as possible from our scWGS data, we applied the KNN imputation method to all datasets used in this study (Supplementary Table S1). The imputed values represented only 0.84%, 0.68% and 1.11% of all copy number values for HeLa S3, JEFF B and MCF-7 cells, respectively. The mean percentage of missing values in the datasets analysed in this study was 0.43% (Supplementary Table S1), suggesting that based on the simulation, the imputed value fidelity was always high (>99%).

### Deep learning model for single-cell DNA replication state classification

Sorting cells by their replication state determined by fluorescence-activated cell sorting (FACS) can induce errors[40]. Therefore, it would be interesting to further validate such replication states by other means. Some computational methods (such as Kronos scRT) can do this validation[22,37,38], but they require manually established thresholds or additional information (e.g. GC content and intra-cellular variability measurements) that is not always directly accessible. To create a method that can bypass the need of metadata, we amalgamated single-cell copy numbers, issued from datasets with replication states inferred from FACS[35], Kronos scRT[37], or their intersection[32,38], depending on the extraction methods, and harvested the labelled replication states to create a deep learning model based only on single-cell DNA copy numbers (Fig. 1c). Data integration resulted in 5,250 replicating and 2,273 non-replating cells from six cell line types (Supplementary

Table S2). We hypothesised that the different ploidy landscapes of the selected cell lines would make this prediction tool universal and adapted to any ploidy state.

We then split our dataset (80:20 ratio) to create a training dataset and a test dataset. To prepare the model to handle noise from lower quality datasets, we applied data augmentation for the training dataset by replicating and artificially altering half of its cells to induce a random noise of ±1 copy number sporadically. We trained the model for copy numbers in 25, 100 and 500 kb bins that resulted in accuracy rates of 97.94%, 98.54% and 98.14% for replication state classification in the test dataset, respectively. Therefore, we selected the 100 kb bin size (best accuracy and spatial resolution) for the downstream analyses. To quantify how well our 100 kb model performed compared with FACS, we calculated the discordance percentage between our in silico predictions and the FACS metadata of three cell types: wild-type (WT) HCT-116 colon cancer cells (n = 713), HCT-116 double knockout (DKO1) cells in which both the maintenance DNA methyltransferase DNMT1 and the de novo DNA methyltransferase DNMT3B were knocked out (n = 668), and GM12878 B cells (n = 3180). We observed that FACS misclassified 17.67% of WT HCT-116, 27.70% of DKO1 HCT-116, and 25.72% of GM12878 cells (Fig. 2). These results demonstrate

that even when taking into account the 1.56% error rate of our model, it generated results with higher accuracy compared with FACS for cell-phase sorting, in accordance with previous estimations[38].

## Unsupervised machine learning for cancer subpopulation discovery

Our next goal was to detect copy number differences among cells, a crucial factor in cancer emergence and progression. To achieve this, we created a 3-step framework to detect genomic subpopulations, i.e. groups of cells with distinct CNV signatures compared with other cells from the same sample (Fig. 1d–f). The autosomal copy numbers of non-replicating cells, determined by our replication state classifier, underwent dimensionality reduction by uniform manifold approximation and projection (UMAP), to be represented in a two-dimensional (2D) space. Then, we used the 2D cell coordinates in these new representations to detect cell subpopulations using density-based spatial clustering of applications with noise (DBSCAN), an unsupervised spatial clustering algorithm. Although UMAP is relatively stable because it is a stochastic algorithm[41] that can generate non-representative distances of high-dimensional data, we repeated the UMAP/DBSCAN steps another six times using random seeds ranging

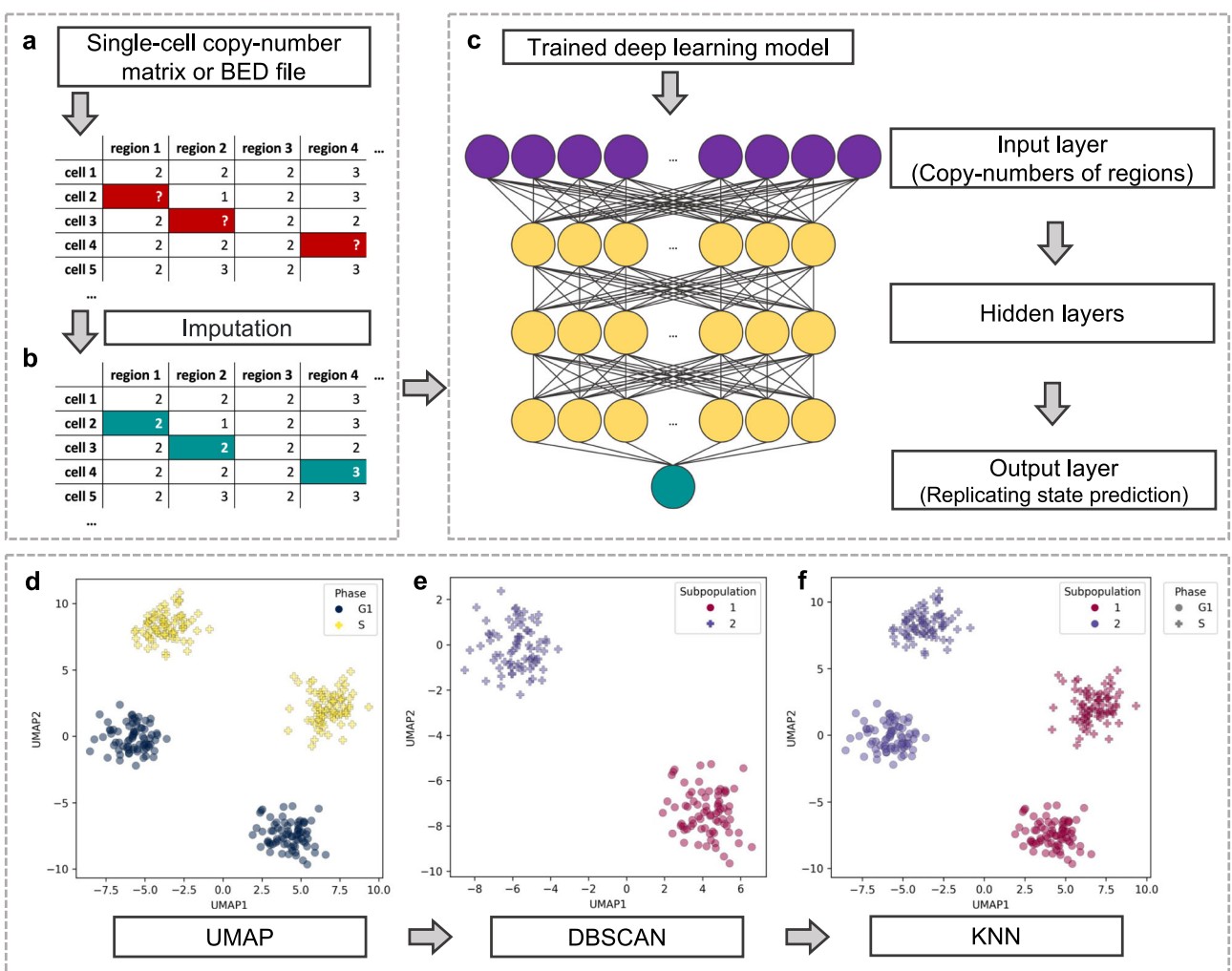

**Fig. 1 | Machine learning techniques used in MnM. a, b** Copy number imputation with k-Nearest Neighbors (KNN). Single-cell copy number data in a matrix or a BED file are used as input and the missing copy number values marked as a question mark (**a**) are filled in by KNN imputation (**b**). **c** Deep learning model for the single-cell replication state classifier. The trained deep learning model includes one input layer and three hidden layers. The output layer is loaded and used to predict the replication states of single cells. **d–f** Subpopulation discovery in three steps. Dimensionality reduction is performed with UMAP and non-replicating cells in two dimensions to provide representative lower dimensions of the copy number data (**d**). DBSCAN clusters the data based on the UMAP coordinates (**e**). This allows matching replicating cells to the corresponding non-replicating subpopulations with KNN after a second 10-dimension UMAP dimensionality reduction step (**f**).

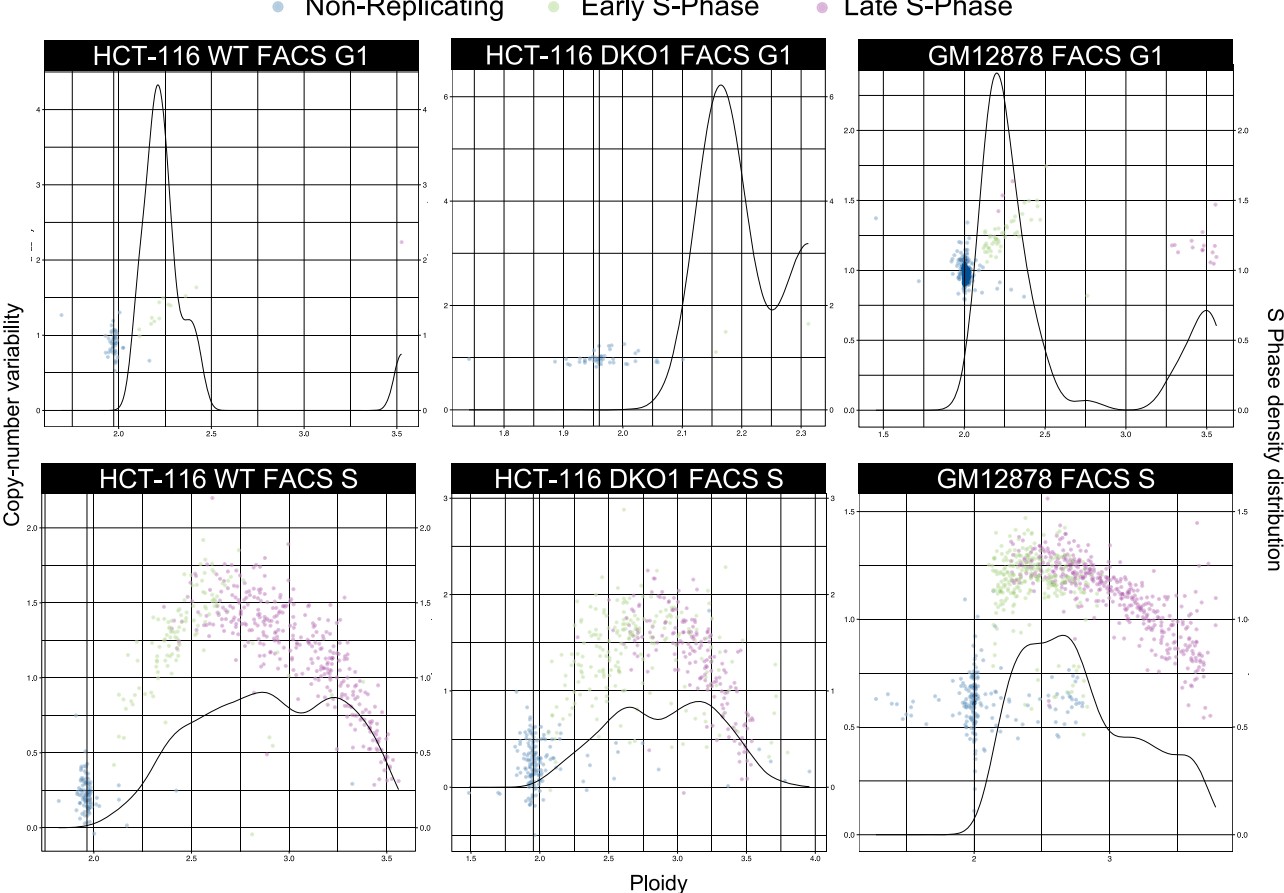

**Fig. 2 | Widespread misclassification of DNA replication states by FACS for the HCT-116 WT, HCT-116 DKO1 and GM12979 cell lines.** Partial discordance for the replication states of single cells between the FACS results and the supervised deep learning method developed here. FACS sorting of G1 cells (upper panels) and of S cells (lower panels) from wild-type (WT; left) and double knock-out of the maintenance DNA methyltransferase DNMT1 and the de novo DNA methyltransferase DNMT3B (DKO1; centre) HCT-116 cell and GM12878 (right) cell samples. Coloured dots indicate the MnM replication state predictions. Source data are provided as a Source Data file.

between 3 and $2^{30}$. The number of sub-populations was counted for each seed. If the predominate number of clusters was not found in the original iteration, the seed would change to the first of the six random seeds that encountered the predominate number of clusters. Sub-populations were iteratively merged when they presented a median copy-number identity >98.5% in a prioritised (decreasing identity) order. Finally, replicating cells were included and a second dimensionality reduction step down to 10 dimensions allowed the KNN algorithm to match replicating cells to their corresponding non-replicating subpopulations.

To validate this method in distinct genome-wide CNV landscapes, we mixed JEFF B ($n$ = 1455 cells) and HeLa ($n$ = 752 cells) cell copy number data and analysed them as if they were a single sample, with the expectation that the two cell lines would be correctly distinguished. We first observed that replicating cells from both cell lines were visually distinguishable in the 2D landscape (Supplementary Fig. S2a–c). After running our 3-step subpopulation detector (Fig. 1d–f), we confirmed that without providing any information on the cell origins, both replicating (Supplementary Fig. S2c) and non-replicating cells (Supplementary Fig. S2d) were matched back into two populations that corresponded to JEFF B and HeLa cells with high accuracy (99.83% and 99.82% for HeLa and JEFF B cells, respectively). Furthermore, we found that female-derived JEFF B cells harboured

only one chromosome X copy (Supplementary Fig. S2d), instead of two copies, a phenomenon compatible with acquired monosomy X.

## Fast and accurate subpopulation discovery and replication analysis

We integrated our machine learning approaches to provide MnM (Mix 'n' Match), a stand-alone and ready-to-use computational tool, which unifies these techniques under one programme. Copy number imputation, replication state classification and subpopulation detection, taken together, allowed scRT extraction from heterogenous cell populations and related downstream analyses, for in vivo and in vitro samples.

We previously reported the identification of two subpopulations of MCF-7 cells, a breast cancer cell-line with unstable aneuploidy[37]. We now used our MnM tool to automatically detect subpopulations from a single-origin MCF-7 cell sample ($n$ = 2768). We identified the two subpopulations based on sub-chromosomal (Fig. 3a) and whole-chromosome (Fig. 3b) copy number differences. These two subpopulations were also separated in the UMAP lower-dimensional space (Fig. 3e). We then used the same method with WT HCT-116 cells and discovered the existence of two subpopulations (Fig. 3f), which was previously unreported[32]. Unlike the MCF-7 cells, these HCT-116 cell subpopulations could only be distinguished at the sub-chromosomal

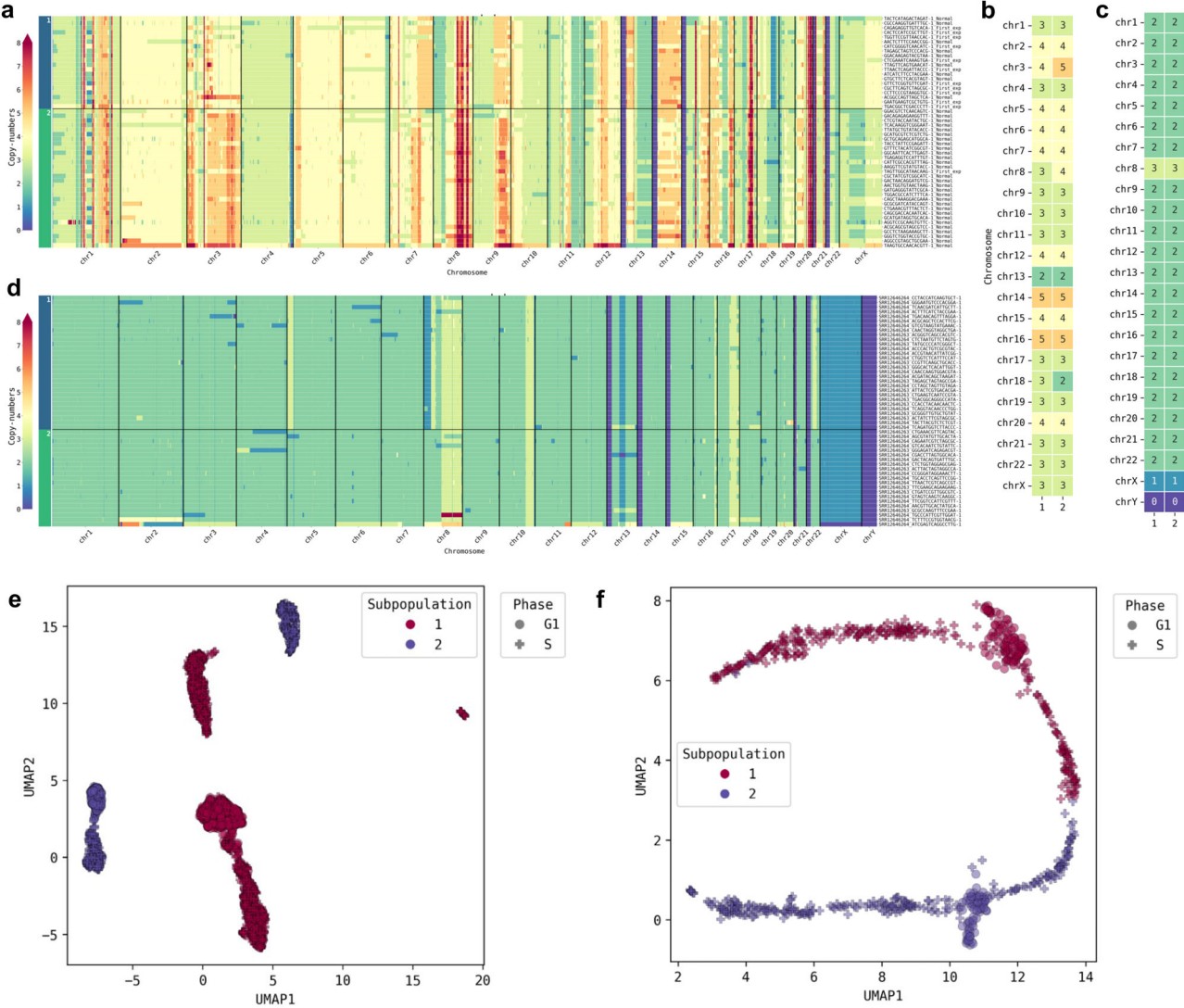

**Fig. 3 | Genomic heterogeneity detected in individual samples of cancer cell lines. a–d** Genome-wide copy numbers (**a, d**) and their median number per chromosome (**b, c**) in MCF-7 (**a, b**) and WT HCT-116 (**c, d**) cells. **e, f** Reduced dimension planes by UMAP showing the subpopulation clustering of MCF-7 (**e**) and WT HCT-116 (**f**) cells based on their copy number profiles. Source data are provided as a Source Data file.

level (Fig. 3c, d), suggesting that the observed local CNV changes were due to DNA repair pathways alterations rather than global genome instability. This is in line with the fact that the HCT-116 cell line harbours a defective mismatch repair pathway (due to a homozygous mutation of the mismatch repair gene *MLH1* on chromosome 3) and exhibits microsatellite instability[42,43], two possible causes of this phenomenon.

Besides its high accuracy, MnM is a fast tool with a runtime of 7 m:22 s to analyse 713 WT HCT-116 cells in 100 kb bins running on a macOS 13.5.2 computer system with six cores of an Intel i5 processor.

To determine whether our subpopulation discovery method can be applied to scWGS data obtained and analysed with different techniques, we reanalysed published copy number data of 43,106 cells processed in ref. 26, with many originating from ref. 22. These data were aligned to the hg19 human genome and generated with HMMCopy in 500 kb bins, a reference genome and copy-number estimator that were different from those of the data used for training the deep learning model. Upon visual inspection of the single-cell genome-wide copy number heatmaps (Supplementary Fig. S3), we observed that there were copy number signatures specific to different subpopulations. We concluded that our approach is robust and

efficient even when using scWGS data aligned to different reference genomes (hg38 and hg19), obtained with different techniques (10x scCNV solution and DLP+), split in different bin sizes (100 kb and 500 kb), and processed with different copy number calling methods (Kronos scRT and HMMCopy).

## DNA replication timing retains high fidelity despite CNVs

We then split the copy number data by subpopulation and provided the detected cell phases to Kronos scRT to obtain the RT profiles. As the calculated copy numbers were relative, copy numbers of MCF-7 (Fig. 4a, b) and HCT-116 (Fig. 4c, d) cells in early and late S phase were corrected in 200 kb bins. We observed that the S/G1-phase borderline was non-linear on the bin-to-bin variability scale (Fig. 4a–d). This signifies that the separation of the replication states with previous computational methods using linear techniques and a unique cut-off[37,38] introduced a larger error rate. For each subpopulation, we inferred and visualised the scRT profiles (Fig. 4e, f). Despite the presence of genome-wide CNVs, the pseudo-bulk RT profiles of the two MCF-7 subpopulations had a Spearman correlation of 93.6% (Fig. 4g) and were also highly correlated with the bulk RT profile (Spearman correlations of 92.8% and 94.4%, respectively). As expected, due to the

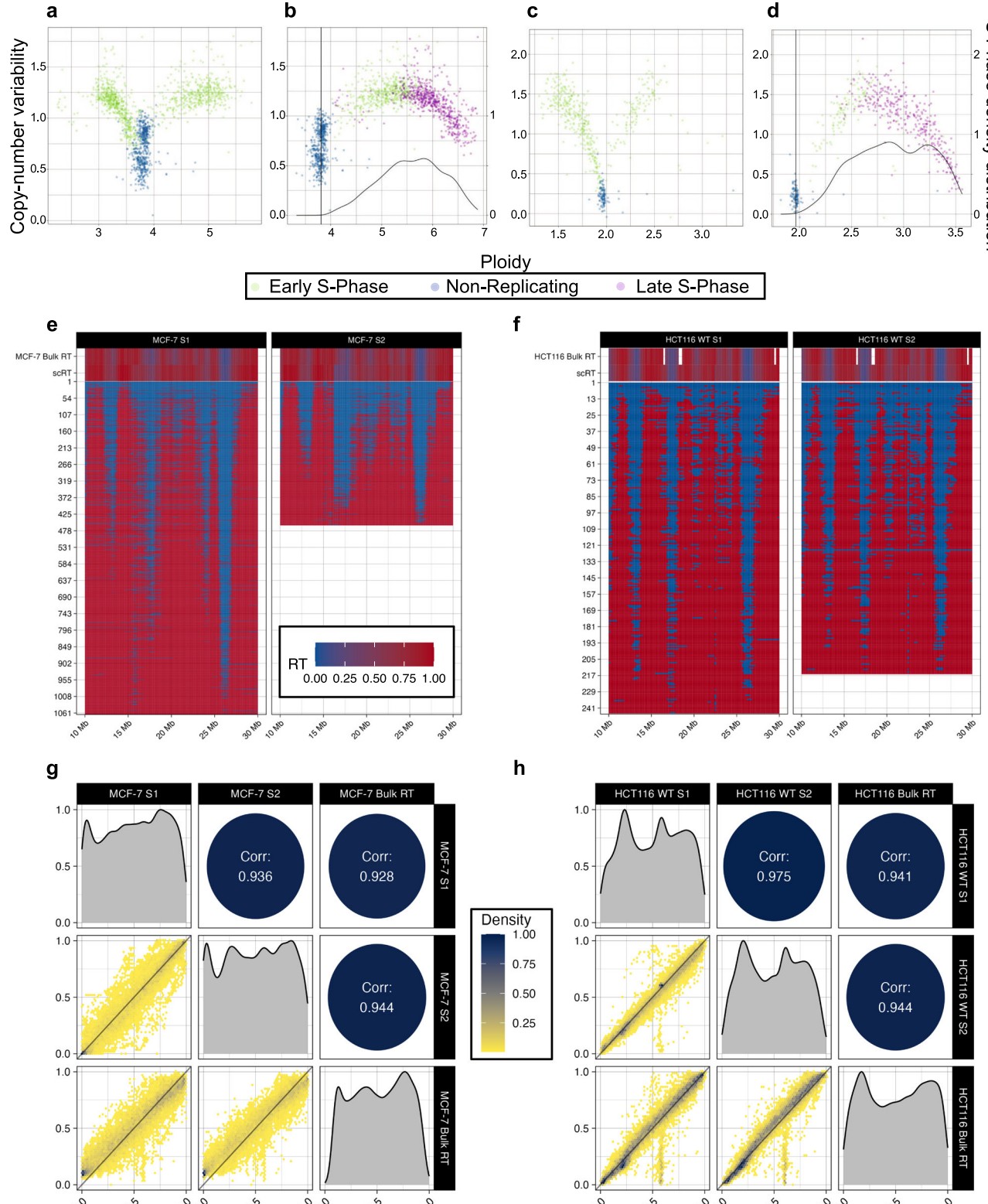

**Fig. 4 | Uncovering the single-cell Replication Timing (scRT) of heterogenous cancer cell lines. a–d** Early (green) and late (purple) S-phase cells from MCF-7 (**a**, **b**) and S phase FACS-predicted WT HCT-116 (**c**, **d**) populations are corrected (**b**, **d**) from raw scCNV data (**a**, **c**) that display non-replicating (blue) and replicating (green) cells. **e**, **f** Chromosome 16 scRT landscapes of MCF-7 (**e**) and WT HCT-116 (**f**) cells display minor differences between subpopulations. Bulk replication timing (RT) and pseudo-bulk RT data are displayed in the upper window. Correlations between pseudo-bulk subpopulation scRT and bulk RT data for MCF-7 (**g**) and WT HCT-116 (**h**) cells. Source data are provided as a Source Data file.

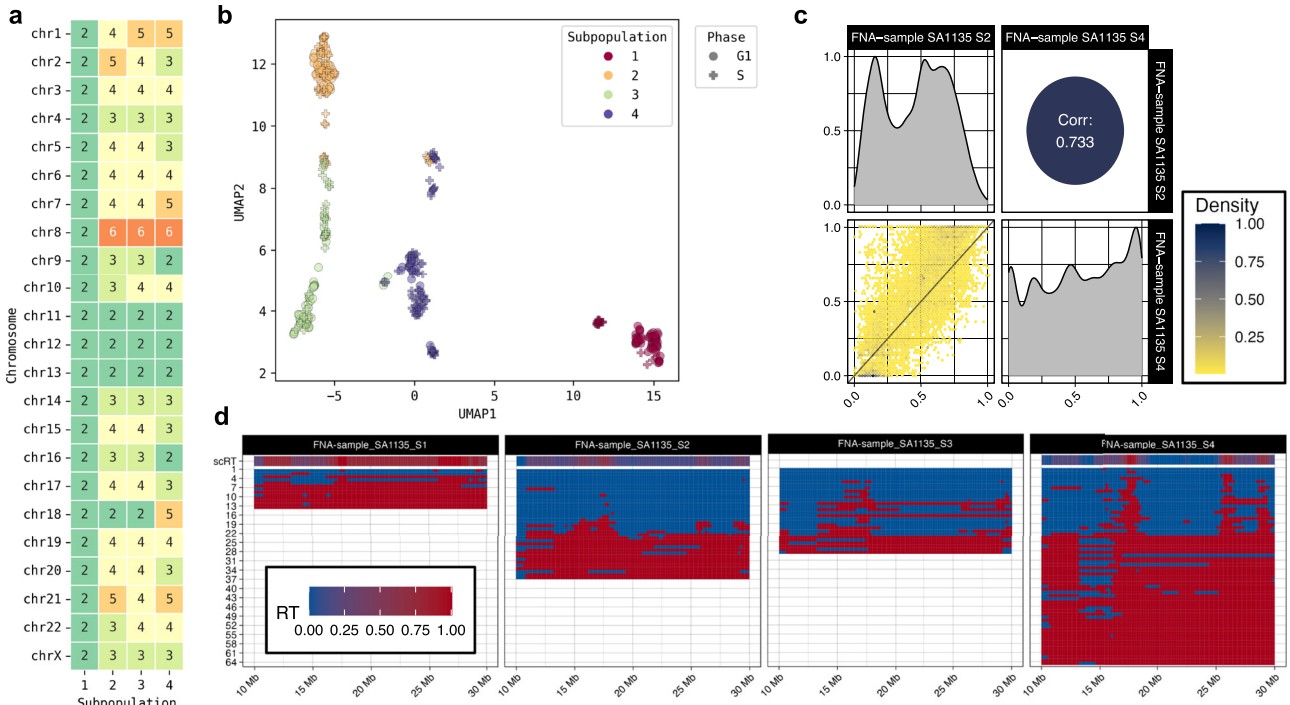

**Fig. 5 | Unravelling the scRT data of cell subpopulations from the human triple-negative breast cancer sample SA1135. a** Major differences in the median DNA copy numbers per chromosome for the four subpopulations. **b** Reduced dimension UMAP plane of the copy numbers of the single cells from the tumour sample. **c** Spearman correlation between the subpopulations 3 and 4. **d** Chromosome 21 scRT landscapes in the four subpopulations, with the pseudo-bulk replication timing (RT) data from the scRT profiles in the upper windows. FNA: Fine-Needle Aspiration (cell collection technique). Source data are provided as a Source Data file.

smaller copy number signatures, the HCT-116 cell profiles were also highly correlated with a Spearman correlation of 97.5% (Fig. 4h).

**Replication timing heterogeneity in a patient tumour sample**
As RT has not been studied in heterogenous tumours, we used the same methods to discover cell phases and subpopulations using published data obtained from a triple-negative breast cancer (TNBC) tumour sample (SA1135). As in the original study[22], we discovered one diploid and three aneuploid subpopulations in non-replicating cells ($n = 58, 54, 55, 26$, respectively; Fig. 5a, b). Among the 347 cells that passed the quality control, 152 were replicating; this proportion of replicating cells was larger than in the cell line models, in agreement with the persistent proliferation of TNBC cells. We then calculated the scRT profiles for each subpopulation. We considered that subpopulations 1 ($n = 13$ cells) and 3 ($n = 29$ cells) did not have a representative S-phase landscape (Fig. 5d) and disregarded them. On the other hand, subpopulations 2 ($n = 36$ cells) and 4 ($n = 74$ cells) showed distinct RT programmes, indicating a deregulated replication programme in vivo. The Spearman correlation value for the RT profiles of subpopulation 2 and 4 from the same tumour was 73.3% (Fig. 5c), which is close to the previously reported correlation coefficient range (Pearson r: 0.76−0.86) for RT profiles of cell lines from different breast cancer subtypes[44]. Our results demonstrate that RT profiles can be very heterogeneous in cell subpopulations from the same cancer sample.

**The scRT atlas reveals cell type and tumour-specific relationships**
We then tested our methods using other datasets. In total, we analysed the copy numbers of 119,991 quality-controlled cells originating from 92 different samples spanning across 21 different cancer cell lines, 35 patient tumour samples and 19 patient-derived xenograft (PDX) samples (Supplementary Table S3). In some cases, we discovered large copy number differences in the same cell line using data obtained from

different publications. The HeLa and MCF-7 cell lines displayed a different karyotype depending on their origin, suggesting that extensive culture of cancer cell lines can induce important copy number changes (Supplementary Fig. S3). It is unlikely that these differences were the result of the different scWGS techniques used for data generation or of confounding batch effects because the HCT-116 cells from two different publications[32,38] had the same karyotype. Importantly, the observed results from HeLa and MCF-7 cells are in line with previous studies reporting karyotypic heterogeneity among and within different HeLa[45] and MCF-7 strains[46]. In addition, the presence of these two MCF-7 cell subpopulations was confirmed experimentally by FISH[37].

We computed the pseudo-bulk RT profiles when we had enough cells to reconstruct a representative S-phase landscape. This was determined either by software failure or visual inspection of the ploidy/copy-number variability plots (i.e. Fig. 2) to make sure that cells were either in mid-S phase or in early and late S phase. When this was not the case, we did not consider the (sub)populations for RT profile generation. We used 41 (sub)populations and calculated the Spearman correlation for each pair (Fig. 6). Unlike the two subpopulations from the same MCF-7 cell sample, we noticed that the RT profiles of MCF-7 samples from different laboratories only presented a Spearman correlation of 84.5%, on average. As this cell line has a variable karyotype, we speculated that these RT profile differences could be caused by widespread copy number differences. Indeed, we discovered that, in MCF-7 cells from different laboratories, the median copy number per chromosome was the same only in 11 and 13 chromosomes (in function of the MCF-7 cell subpopulation; Fig. 4, Supplementary Fig. S3). On the other hand, the RT profiles of JEFF B and GM lymphoblastoid cell lines displayed high Spearman correlation coefficients, all >90%, regardless of the sample origin. Although both H7 human embryonic stem cells (hESCs) and GM12892 lymphoblastoid cells presented a perfectly diploid karyotype (Supplementary Fig. S3), the Spearman correlation coefficients for their RT profiles were lower (79%), illustrating that in

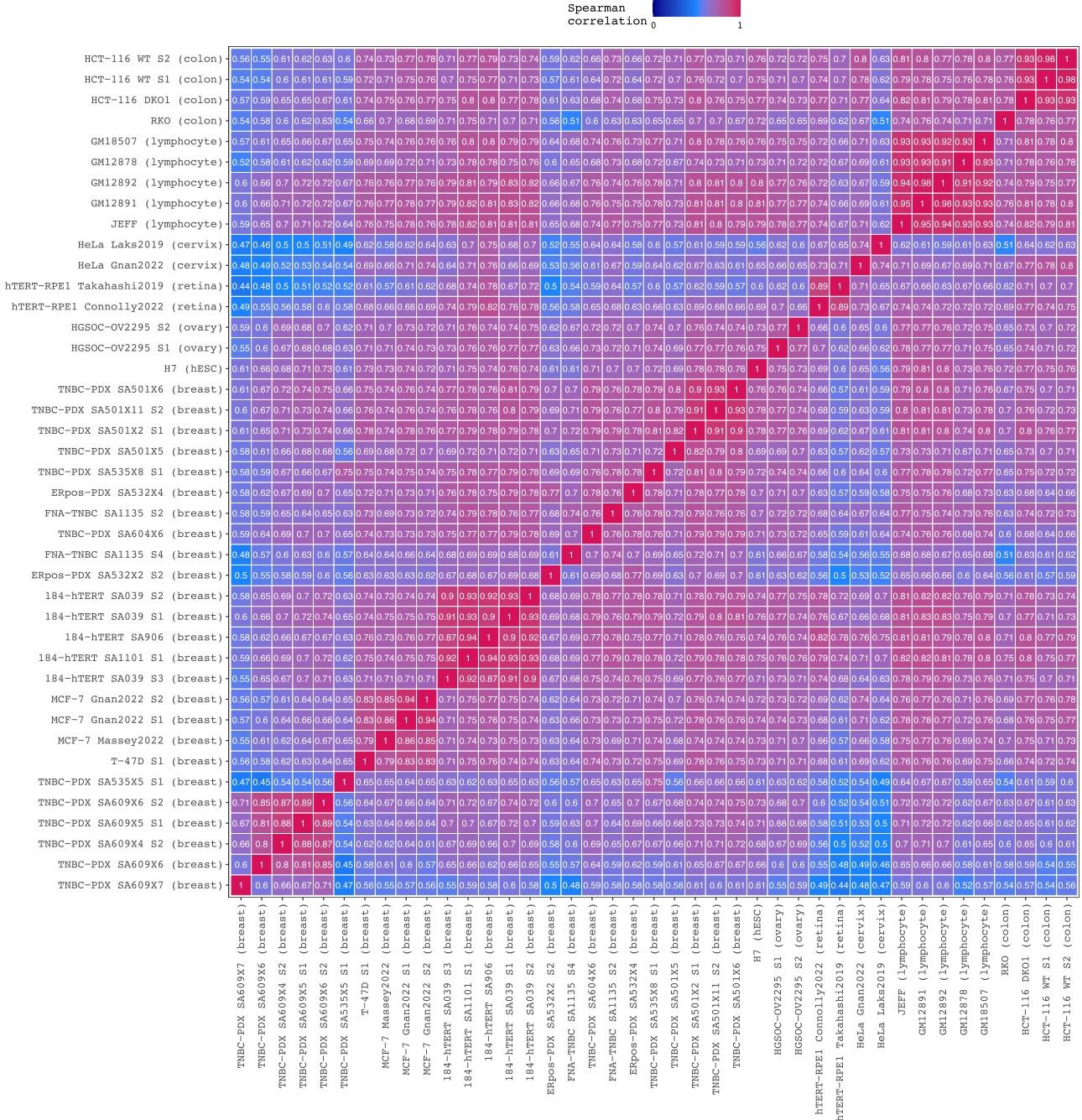

**Fig. 6 | The scRT atlas reveals cell-type specific relationships of DNA replication.** Spearman correlations between the 41 scRT pseudo-bulk profiles extracted from human cell lines, patient tumours and patient-derived xenograft (PDX) samples. Samples were ordered by hierarchical clustering using the second version of Ward's minimum variance method (Ward.D2). Cell subpopulations from the same sample are indicated with an S after the sample name (e.g. S1, S2). Cell origins (tissue) are given in brackets after the sample name. PDX samples are identified as

sample_numberXmouse_passage. hESC: human embryonic stem cells; RPE: retinal pigment epithelial cells; TNBC: triple-negative breast cancer; ERpos: oestrogen receptor positive; PDX: patient-derived xenograft; HGSOC: high-grade serous ovarian carcinoma; hTERT: human telomerase reverse transcriptase. FNA: fine-needle aspiration (cell collection technique). Source data are provided as a Source Data file.

addition to CNVs, scRT profiling can be used as a cell-type specific biomarker, which can help to determine the potential origin of tumour cells.

To better examine the similarities and differences of the various RT profiles of the scRT atlas, we generated RT trajectories using potential of heat-diffusion for affinity-based trajectory embedding (PHATE)[47] to visualise the local and also the global structure of the RT landscapes. Initially, we generated the RT trajectories in the well-characterised cell lines of this study (Fig. 7a, c) and found distinct RT

dynamics for different cell types, consistent with previous findings[37]. Notably, the trajectories of the lymphocyte lines (i.e. JEFF, GM12891, G12892) and HCT-116 cell subpopulations reflected the closeness of correlations observed in the scRT atlas (Fig. 6), whereas MCF-7 cell subpopulations showed slight divergences, as previously reported[37]. We then extended this analysis to the breast cancer samples and PDXs of the atlas (Fig. 7b, d). We observed a general conservation of RT profiles by tumour; however, we also detected RT profile divergences during PDX passaging (e.g. TNBC PDX SA609), reflecting RT profile

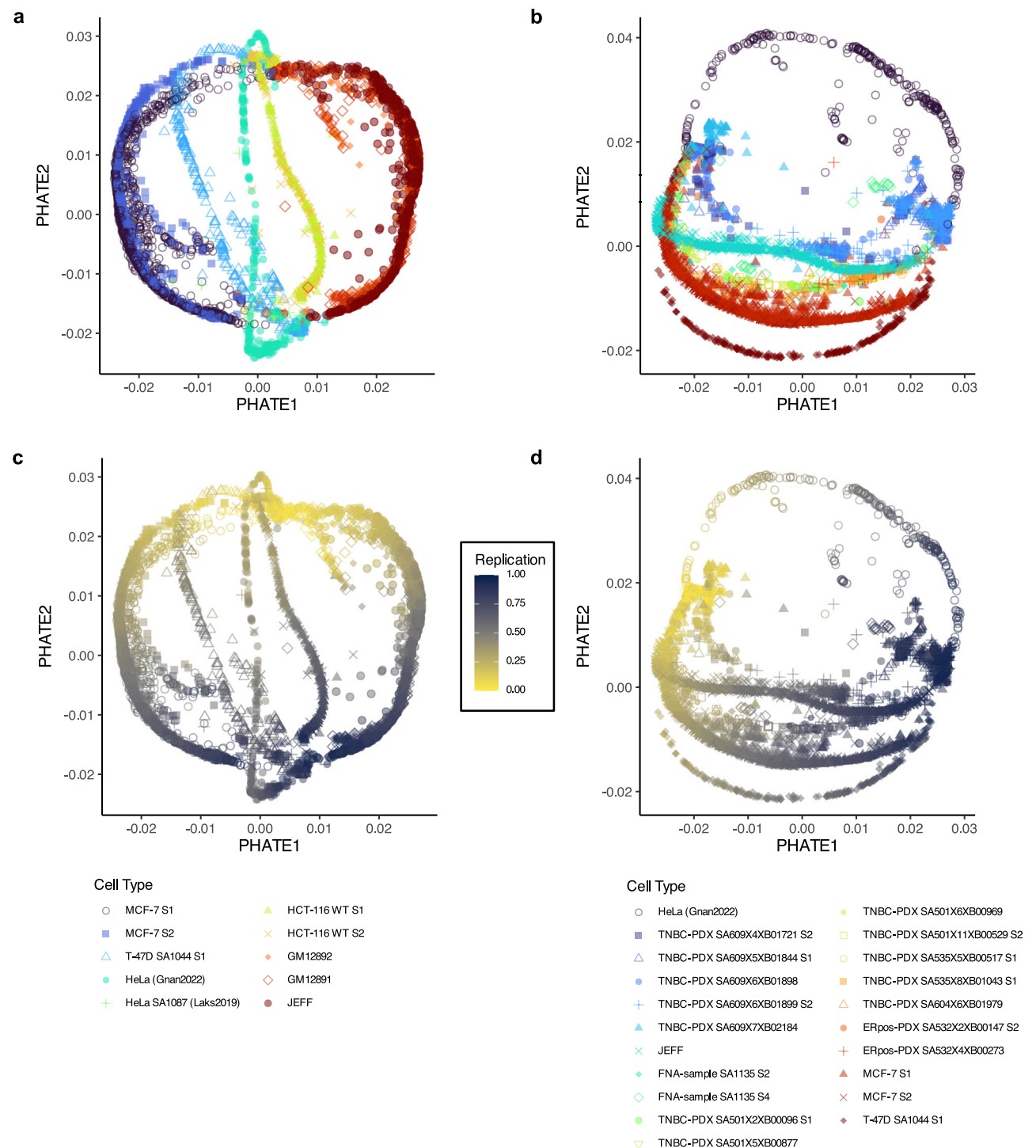

**Fig. 7 | The scRT trajectories reflect relationships between cell types and cell subpopulations. a** scRT trajectories of different cell lines; each point represents a single cell (colour and shape in function of the cell type). **b** scRT trajectories of breast cancer samples, including PDX, Fine-Needle Aspiration (FNA) samples, and cell lines. Data of representative cell lines are included for comparison. **c** scRT trajectories of different cell lines, coloured according to the percentage of cell replication (DNA content replicated). **d** scRT trajectories of breast cancer samples, coloured according to the percentage of cell replication. Cell type colours and shapes are indicated in the internal figure legends corresponding to (**a, c**) and (**b, d**), respectively. Source data are provided as a Source Data file.

evolution during cancer progression. Notably, the trajectories uncovered subtle shifts in RT patterns, suggesting that RT profiling can capture genomic changes during tumour progression.

To assess whether there was a significant difference in RT trajectories among subpopulations, we collected the scRT trajectory coordinates from Fig. 7b, grouped by subpopulation and used them for a permutation test (see Methods), a non-parametric method that does

not assume specific data distributions and allows the randomisation of group (i.e. cell-type/subpopulation) labels. In agreement with our previous report[37], the trajectories of the two MCF-7 cell subpopulations were significantly different ($p = 0.004$). We generated pairwise $p$-values for each (sub)population (Supplementary Fig. S4) and demonstrated that the RT trajectories of some tumour samples (e.g. TNBC PDX SA501 and PDX SA532, ER-positive breast cancer) did not change

through PDX passaging, while in other samples (e.g. TNBC PDX SA609) they constantly changed. Furthermore, the breast cancer sample SA1135 showed an RT trajectory change between subpopulations. Altogether, these results show that RT subpopulation extraction can be used to study RT dynamics during cancer progression in vitro (i.e. cell lines) and in vivo or ex vivo (i.e. tumour/PDX samples).

## Discussion

In this work, we present a machine learning-based approach to identify single-cell replication states and genomic subpopulations in single heterogenous samples, shedding light on an additional layer of heterogeneity in cancer progression. Our tool, MnM, is designed to accurately establish single-cell replication states and identify genomic subpopulations based on the DNA copy numbers of a mixture of heterogenous cells issued from a single sample. By leveraging this information, MnM disentangles heterogeneity, aligning replicating and non-replicating cells in each subpopulation. The validation of each MnM step demonstrated remarkable accuracy in missing copy number imputation and cell replication state classification. Considering that the data used in our current study present a low rate of missing copy number values (median = 0.34% in the present study), the KNN approach used here is robust enough for imputation without a high computational cost. For more sparse data, other deep learning approaches (such as Generative Adversarial Nets)[48,49] could also be considered. We also confirmed that with subpopulation clustering, MnM can efficiently detect the number of different cell types and as well as the underlying subpopulations in a single sample.

Our study underlines the FACS limitations for cell cycle phase detection (at least for the scWGS data with available FACS sorting data included in the current study), revealing error rates that ranged from 17.7% to 27.7%, in accordance with previous estimations[38]. Although in some cases single cells are sorted into the different cell cycle phases before sequencing, and the latest version of cell sorting machines/methods can provide more accurate results, we believe that cell-sorted metadata should always be verified computationally to avoid erroneous conclusions. Our replication state classifier was only trained on the hg38 reference genome; however, this method can easily be extended to other genomes and used routinely. Furthermore, in some cases, FACS metadata may not exists (e.g. unsorted samples), such as in the case of hESCs and tumour samples that contain a limited number of cells that one cannot afford to lose in additional analyses (e.g. FACS). Therefore, in silico predictions are valuable for these cases. Nonetheless, we acknowledge that the amount of data used here to compare our model is limited and as a result, it would be interesting to further test our model using new scWGS data as they become available.

Another noteworthy finding was the identification of chromosomal aberrations in most samples. We discovered that the JEFF B cell line had lost one copy of the chromosome X, a phenomenon correlated with mitotic errors occurring during ageing[50]. We also detected important chromosomal aberrations in cell lines and tumour samples (Supplementary Fig. S3), further underlining the importance of DNA copy number screening. Aneuploidy is an omnipresent trait in tumour genomes[10,51]. The presence of genomic instability in cancer has been recognised for a long time[52–54]; however, its exact role in tumour development remains ambiguous. In fact, the omnipresence of aneuploidy in tumours prompts discussion on its dual role as a tumour promoter and suppressor[55,56]. As discussed by others[10], copy number gains might amplify the expression of tumour-promoting genes (i.e. oncogenes) or might stem from the disruption of cell cycle checkpoints, a common occurrence in advanced malignancies[57,58]. In specific cases, aneuploidy might surprisingly exert tumour-suppressive effects. For example, genomic instability could decrease tumour fitness[56,59,60] and individuals with Down syndrome, arising from the triplication of chromosome 21, are less susceptible to developing solid tumours[61].

The robustness of the DNA replication machinery is a cornerstone of cellular integrity. Its deregulation can lead to genomic instability, a cancer hallmark. Many pioneering studies revealed the important links between the DNA RT programme and cancer mutation rates, signatures and structural variants[9,44,62–64]. However, due to the lack of RT data directly measured in tumour samples and despite a general association observed between the RT programme and genetic alterations in cancer, few studies have been able to analyse how the RT programme plasticity and changes interact with the mutational landscape in cancer and contribute to cancer development. In a recent study[44], Dietzen and colleagues focused on RT alterations by comparing RT data from lung and breast cancer cell lines and matched normal cell lines. They showed that RT is altered relatively early during cancer development/progression and that this plays an important role in shaping the cancer mutational landscape. Interestingly, they found that regions changing from late to early replication contain genes that are overexpressed in cancer and present a preponderance of APOBEC3-mediated mutation clusters associated with driver mutations. These results suggest a direct role of RT in tumour development/progression. In the present study, we observed a remarkable contrast between cell models and patient-derived samples in terms of RT alterations. Cell models exhibited relatively modest changes in DNA replication dynamics. These cell lines, cultured in controlled conditions, often are simplified representations of cellular systems. Conversely, in patient tumour samples, we identified substantial disruptions in DNA replication patterns. These observations resonate with the importance of the tumour microenvironment and its intricate interplay with replication stress and thus genomic stability. The RT dynamic disparities between cell lines and patient samples highlight the need of integrating complex, patient-specific factors to understand DNA replication mechanisms in the context of cancer progression. Until now, only RT data from cancer cell lines were available and previous studies neglected the intra- and inter-tumour heterogeneity. Here, we provide a tool that allows generating unique data on subpopulation-resolved RT profiles from tumour samples. These results show that this tool can facilitate the exploration of the intra-tumour RT programme heterogeneity and its relationship with genomic alterations in individual patient tumours.

Despite advances in scWGS studies, there are still challenges in achieving complete genome coverage due to the low read depth. As new methods are emerging, notably a recent report of long-read single-cell sequencing[65], future studies will be able to thoroughly investigate the precise relationship between mutational landscape, aneuploidy and replication programme. Eventually, the imminent generation of higher resolution scRT data will allow addressing the RT differences in homologous chromosomes. This underlines the need of detailed analyses of the replication synchronicity of alleles, an even more complex task for aneuploid polyallelic cancer cells.

In conclusion, we developed a machine learning-based tool (MnM) to democratise scRT subpopulation detection from DNA copy numbers and also built a large scRT atlas for the community. This could be a valuable resource for future research. This tool can contribute to our understanding of cancer emergence and progression. For instance, this tool allowed obtaining evidence of a further layer of widespread heterogeneity in cancer progression, based on the DNA replication patterns. Lastly, our results underline the necessity to consider tumour samples in order to fully understand the mechanisms governing DNA replication during cancer development/progression. Indeed, cancer cell lines are an easier research model, but they lack some critical environmental factors that contribute to cancer development.

## Methods

### MnM: mix 'n' match single cells

MnM consists of three main steps: passing single-cell copy number data through (i) KNN imputation for missing copy number values, (ii) a supervised replication state classifier (to separate replicating from non-replicating cells), and then (iii) an unsupervised subpopulation detector (see below for details). The programme can load a BED file containing DNA copy numbers for each genomic region (obtained with scWGS data) with cell indices, or alternatively, a matrix with genomic regions as headers (chr:start-end) and individual cell identifiers as row indices (Fig. 1a). Missing copy numbers are filled-in with KNN imputations (Fig. 1b), and the cell replication state is predicted using a pre-trained and ready-to-use deep learning model (Fig. 1c, d). Then, copy numbers of non-replicating cells undergo a lower dimensional space transformation to discover underlying subpopulations by unsupervised clustering (Fig. 1e). Lastly, replicating cells are matched with their corresponding non-replicating populations in a dimensionality-reduced landscape (Fig. 1f) for further analyses, such as scRT extraction or identification of the different CNV subpopulations in the non-replicating cells. A schematic representation of MnM main steps (Fig. 1) illustrates that the combination of deep learning, UMAP, DBSCAN and KNN algorithms allows uncovering replication states and subpopulations from single-cell whole-genome copy number calling data (detailed in the following sections).

### scWGS demultiplexing

The BAM files of hg19-aligned GM12878 B-cell data from ref. [38]. were obtained, sorted by read name with samtools[66] sort (v. 1.16.1; option -n) and converted to fastq files with samtools fastq (options -T CB --barcode-tag CB) to transcribe the barcodes to the fastq headers from the BAM headers for alignment to the hg38 reference genome, as indicated below. These files were demultiplexed with demultiplex[67] demux (v 1.2.2; options -m 0 --format x). Data for other cell types obtained from the 10X Chromium Single Cell CNV solution from ref. [28,32,38]. (Supplementary Table S3) were acquired as fastq files and demultiplexed using demultiplex demux (options -r -e 16) with barcodes extracted from the first 16 bp of forward reads. All extracted barcodes were filtered based on the 10X barcode whitelist.

### Barcode validation

10X single-cell data were considered for the analysis if they originated from valid barcodes that were identified as follows. Data were prepared by counting the number of lines of each demultiplexed fastq file and then divided by 4 to reflect the number of total reads per single cell. The resulting list containing the number of reads per barcode was then used to separate corrupted or low-read (invalid) barcodes from qualitative (valid) barcodes using a custom R[68] (v4.0.4) script (see 'Code Availability'). Barcodes containing less than 30,000 reads were not considered qualitative due to the very low number of reads and were systematically removed to eliminate any noise in the initial peak and with the goal of only keeping a mixture of two distributions (valid and invalid barcodes). The em command from the cutoff R library (v0.1.0) was used to identify the cut-off point of 2 log-normal distributions of the read counts from the Expectation-Maximisation (EM) algorithm for each demultiplexed file.

EM included two stages: the expectation step (E-step) and the maximisation step (M-step), which occurred after initialisation of the μ and σ parameters (see below) for the 2 log-normal distributions (D1 and D2). The probability density function (PDF) used in the E-step (formula 1), which represented the probability of observing a particular read count per barcode (continuous random variable) given the following parameters, of the log-normal distribution can be described as:

$$f(x|\mu, \sigma) = \frac{1}{x\sigma\sqrt{2\pi}} e^{-(\ln x - \mu)^2/(2\sigma^2)} \tag{1}$$

where:
- $x$ represents the read count of the barcode,
- $\mu$ represents the mean (also called location parameter),
- $\sigma$ represents the standard deviation (also called the scale parameter).

Using this, $\gamma_i$ (formula 2), which represents the probability that barcode read count $i$ belongs to the valid distribution was calculated as:

$$\gamma_i = \frac{f(x_i|\mu_{D2}, \sigma_{D2})}{f(x_i|\mu_{D2}, \sigma_{D2}) + f(x_i|\mu_{D1}, \sigma_{D1})} \tag{2}$$

where:
- $f(x_i|\mu_{D2}, \sigma_{D2})$ is the PDF of the log-normal distribution with parameters $\mu_{D2}, \sigma_{D2}$ evaluated at the read count $x_i$ for the valid distribution (D2),
- $f(x_i|\mu_{D1}, \sigma_{D1})$ is the PDF of the log-normal distribution with parameters $\mu_{D1}, \sigma_{D1}$ evaluated at the read count $x_i$ for the invalid distribution (D1).

The E-step computed the expected value of any missing data points and calculated the probabilities of the missing or overlapping data given the estimates of μ (formula 3) and σ (formula 4). Then, the M-step updated the parameters of the log-normal distributions using the estimated probabilities as follows:

$$\mu_{new} = \frac{\sum_{i=1}^{N} \gamma_i \ln x_i}{\sum_{i=1}^{N} \gamma_i} \tag{3}$$

$$\sigma_{new} = \sqrt{\frac{\sum_{i=1}^{N} \gamma_i (\ln x_i - \mu_{new})^2}{\sum_{i=1}^{N} \gamma_i}} \tag{4}$$

The E- and M-steps were repeated iteratively until the estimated probabilities $\gamma_i$ converged (when the parameters and probabilities stopped changing between iterations). The exact cut-off value between D1 and D2 was obtained with the cutoff command from the same package, with D1, the lower read count distribution, belonging to the Type-I error. Only the barcodes with a number of reads superior or equal to the EM cut-off value were considered valid and those with a lower read number were discarded. Histograms containing representations of the read counts and cut-off values were systematically generated for visual inspection and validation. The valid barcodes were retained with their respective reads as fastq files that corresponded to the single cells used for mapping and additional analyses.

### Read mapping

The scWGS data of MCF-7 breast cancer[37], JEFF B[37], HeLa S3 cervical cancer[37], and hTERT-RPE1 retinal pigment epithelial cells[35] were aligned to the UCSC human reference genome hg38, as previously reported[37], using the Kronos FastqToBam module. The reads of other single-cell fastq files were trimmed and filtered by quality score with Trim Galore[69] (v0.6.4; options –fastqc, –gzip, --paired when paired-end data were used or omitted otherwise, and --clip_R1 16 except for GM12878 cells that were originally aligned to the hg19 genome) based on Cutadapt[70] (v3.7) and FastQC[71] (v0.11.9) and mapped to the UCSC hg38 reference genome with BWA mem[72] (v0.7.17; option -M). Mate coordinates were corrected using samtools fixmate (option -O bam) when

data were from paired-end sequencing. Then, all BAM files were sorted by coordinates with samtools sort (-O bam) before read duplicate removal with Picard[73] MarkDuplicates (v2.26.11; options ASSUME_SORT_ORDER=coordinate, METRICS_FILE) via java (v19; options -Xmx16g -jar). MultiQC[74] (v1.10.1) was used to visually inspect the single-cell data quality.

## Copy number matrix organisation

Copy numbers from the resulting single-cell BAM files were estimated with the Kronos scRT Binning and CNV commands in 20 or 25 kb windows (see code for details). Problematic genomic regions were masked with the v2 hg38 ENCODE blacklist[75]. The resulting copy number BED files were regrouped by sample and used as input for further analyses and visual representations with MnM and the random seed set to 18671107.

Genomic regions from all MnM input files were rearranged in 100 kb non-overlapping genomic windows (as a median of the copy numbers from the input file that overlapped with the 100 kb window by at least 50%) delimited by the chromosome sizes of the hg38 reference genome provided by bedtools[76,77] and in 25 kb and 500 kb replication state classifier models. Then, MnM automatically processed the data by temporarily removing windows containing no data. Any remaining sporadic missing values were filled in with the integrated sklearn KNN imputation algorithm[39] (options n_neighbors = 5, weights = 'distance'). The nearest neighbours were defined as the five closest cells based on the Euclidean distance of the genome-wide copy numbers (distances calculated in pairs for genomic regions that were not missing in both cells). A weighted average of copy numbers from the region of the closest neighbours was used as the imputation value. The imputation method can be described as follows:

$$\hat{X}_{ij} = \frac{\sum_{k=1}^{n_{neighbours}} w_{ik} \cdot X_{kj}}{\sum_{k=1}^{n_{neighbours}} w_{ik}} \tag{5}$$

where:
- $\hat{X}_{ij}$ (formula 5) represents the imputed value for the copy number of the region $j$ in cell $i$,
- $X_{kj}$ denotes the value of region $j$ in the $k$-th neighbour,
- $n_{neighbours}$ is the number of nearest neighbours considered for imputation. Here $n$=5,
- $w_{ik}$ represents the weight assigned to the $k$-th neighbour for cell $i$ based on the Euclidean distance.

This imputation method was also used for the imputation of 5–55%, in intervals of 5%, of single-cell copy number values that were randomly selected and removed after the elimination of any windows containing missing values of an S-phase-enriched MCF-7, HeLa S3 or JEFF B cell population[37]. A random imputation method (where each missing copy number value was substituted by a randomly selected non-missing value from the matrix) and a median imputation method (where the median of each genomic region was imputed) were implemented for comparison with the KNN imputation method under the same random seed (see code for details). Accuracy was calculated as the percentage of identity of the imputed values compared with the original values. Similarity was calculated as the percentage of values that differed less than ±1 copy number for KNN imputations compared with the original values. Invariance was calculated matrix-wide as the percentage of unchanged copy numbers after imputation.

## Replication state classifier

To organise the data for the replication state classifier, cell cycle phases were extracted with Kronos scRT (for HeLa, MCF-7 and JEFF B cells[37]), only from the FACS metadata (for hTERT-RPE1 cells[35]) or from the intersection of common replication states from the FACS metadata and Kronos scRT (for HCT-116 colon cancer and sorted GM12878

cells[32,38]; Supplementary Table S2). The resulting single-cell copy number matrices were concatenated. Partially or completely missing regions (i.e. any genomic region containing at least one missing copy number value) were removed and only autosomal data were retained. Eighty percent of the cells were used as training data and the remaining 20% were used as testing data. To allow the replication state classifier to distinguish noisy copy number profiles of non-replicating cells (e.g. from low-quality cells or technical noise) from those of replicating cells, data augmentation was performed as follows to reduce overfitting. Half of the CNVs from cells in the training dataset were randomly selected and copied. For each of these copied cell CNV profiles, noise was induced by altering the copy numbers by ±1 in 5–75% of the genomic regions selected from a uniform distribution.

The replication state classifier was built using a Sequential architecture, a feed-forward neural network. The model was designed with the Keras[78] Python library (v2.13.1) to facilitate the construction of a linear stack of neural network layers, each connected to the next one. The single-cell copy number matrix of the training dataset, which contained the six cell types and the simulated data, was used as input. The sequence of layers for hierarchical feature extraction and predictive modelling consisted of three hidden layers with 64, 32 and 16 units, respectively. These layers facilitated the extraction of increasingly complex and abstract representations of the input copy number profiles. The model terminated in an output node with a single unit, using a sigmoid activation. This configuration was suitable for binary classification tasks, enabling the model to produce a probability estimation in a [0,1] range. Upon construction, the model was compiled with a binary cross-entropy loss function to optimise the network performance concerning binary classification. An 'adam' optimiser, which is efficient and adaptive on learning rates, was used to optimise the parameters throughout training. To avoid overfitting, an early stopping mechanism was implemented on an epoch-based patience of 15 iterations.

After training completion, the resulting neural network model and the list of genomic windows comprised in the matrix were saved. Then, the model was integrated and automatically loaded with MnM to predict the single-cell binary replication states (Replicating/S-Phase, Non-Replicating) of the scWGS data from cell lines, patient tumours and PDX samples. When a region required by the model was not present, MnM compensated for the missing values by using linear interpolation from both directions. Compensating for these missing values ensured the continuity and integrity of replication state predictions.

## Subpopulation discovery

Starting with non-replicating cells, the number of variables was reduced from the number of autosomal regions to two dimensions with UMAP[41]. Then, the DBSCAN algorithm[79,80] was used to detect the number of groups in this reduced dataset with min_samples = 10% of the total cell number. The epsilon ($\epsilon$) parameter (formula 6) was calculated as follows:

$$\epsilon = \frac{\max(UMAP1) - \min(UMAP1)}{\max(UMAP2) - \min(UMAP1)} \times 1.25 \tag{6}$$

where UMAP1 and UMAP2 correspond to UMAP first and second output parameters, respectively. Epsilon was always restricted between 1.25 and 2 and at least 10 cells were required to form a subpopulation. UMAP was repeated with six randomly generated seeds and the most frequent number of subpopulations, as defined with DBSCAN, was retained. Subpopulations discovered with DBSCAN were redefined and merged iteratively in a descending similarity order if the median copy numbers per region were 98.5% identical. Copy numbers of both S-phase and non-replicating cells were reduced to ten UMAP dimensions (second UMAP round) and then each S-phase cell was matched to the closest non-replicating group with the sklearn nearest

neighbour command (options n_neighbors = 50% of cells, metric = 'euclidean'). The minimum number of nearest neighbours was five cells. Both UMAP rounds were performed using the single-cell copy number matrices with the addition of five artificial cells stretching from complete haploid to pentaploid profile for subpopulation calibration.

## DNA replication timing

Kronos scRT was modified to be compatible with R v4.0.5, ignore copy number confidence during quality-control filtering, and produce an extra metadata file that contained the cell diagnostic details. The diagnostic module was used at the first stage for quality control based on the number of reads per Mb under developer mode (option -d) created for this purpose. Data were filtered and passed through MnM for replication state classification and subpopulation detection. The Kronos scRT WhoIsWho module was used to assign the cell cycle phases from the replication state classifier or FACS data (see code for details). Then, the diagnostic module was used a second time to correct the early and late S-phase copy numbers (option -C). For each subpopulation and biological replicate, the copy number data were split into different files with a custom Python (v3.9.11) code. Then, Kronos scRT was used to calculate the RT profiles through the RT module in 200 kb windows. The resulting scRT binary values were used to produce scRT trajectories with the DRed module under the random seed '18671107' for reproducibility. Bulk RT profiles were lifted from hg19 to hg38 with the UCSC liftover tool[81] after being converted to bed files with bigwigtobedgraph[82]. RT trajectories were generated in R (v 4.3.3) with PHATE[47] (v1.0.7) using the same methods for distance matrix generation as previously described[37] (using simple matching coefficient distances).

## Statistics

Imputation methods were compared with two-sided paired $t$-tests with the Python (v3.11.8) scipy.stats.ttest_rel function (v1.12.0). Permutation tests on the trajectories were carried out using a custom Python code with 1,000 permutations. The observed test statistic (formula 7) was calculated as the absolute mean of the sum of differences in means between subpopulations for both PHATE coordinates:

$$\text{Observed statistic} = \sum_{\text{groups}} \left| \bar{x}_{\text{group}} - \bar{y}_{\text{group}} \right| \tag{7}$$

where $\bar{x}_{group}$ and $\bar{y}_{group}$ are the means of the PHATE1 and PHATE2 coordinates for each subpopulation, respectively.

The permutation test was executed by randomly shuffling the group labels by keeping the PHATE coordinates intact. The permutation test statistic was computed using the shuffled data in the same way as the observed statistic. This allowed calculating the p-value (formula 8) as follows:

$$\text{Permutation test } p - \text{value} = \frac{\sum_{\text{permuted}}(\text{permuted statistic} \geq \text{observed statistic}) + 1}{\text{number of permutations} + 1} \tag{8}$$

Pseudo-bulk, scRT, and bulk RT correlations were calculated with the Spearman method as previously detailed[37] and scRT correlation clustering for the scRT atlas was ordered with the Ward.D2 hierarchical clustering method.

## Reporting summary

Further information on research design is available in the Nature Portfolio Reporting Summary linked to this article.

## Data availability

The 10x barcode whitelist can be found at https://github.com/TheKorenLab/Single-cell-replication-timing/blob/main/align/10x_barcode_whitelist.txt, the human reference genome hg38 at https://support.illumina.com/sequencing/sequencing_software/igenome.html, and the genomic blacklist at https://github.com/Boyle-Lab/Blacklist. Published scWGS datasets can be found in the Gene Expression Omnibus (GEO) under the accession numbers GSE186173[37], GSE158011[32], GSE108556[35] and in the Sequence Read Archive (SRA) under PRJNA770772[38]. Access to BC Cancer datasets is controlled and requires a data access agreement which can be found at the European Genome-Phenome archive (EGA) under the accession number EGAS00001003190[22]. Processed scCNV data from ref. 26. can be found at https://zenodo.org/record/6998936. Bulk RT profiles were obtained under accession numbers GSE34399 for MCF-7 and GSE158011 for HCT-116 cells. The liftover chain file is available at https://hgdownload.cse.ucsc.edu/goldenpath/hg19/liftOver. Source data can be accessed at https://doi.org/10.5281/zenodo.14260088. The scRT/scCNV atlas generated in this study can be found on MnM's GitHub page [https://github.com/CL-CHEN-Lab/MnM/tree/main/scRT_scCNV_Atlas]. The source data for generating Figs. 2–7 and all Supplementary Figs. are provided as a Source Data file and archived on Zenodo (https://doi.org/10.5281/zenodo.14260088).

## Code availability

The MnM source code is available at https://github.com/CL-CHEN-Lab/MnM under General Terms of License (GTL, Version 2 - 2024/01/02) and archived on Zenodo (https://doi.org/10.5281/zenodo.14261500)[83]. It was registered with the French Agency for the Protection of Programs (APP) under registration number N° IDDN.FR.001.340005.000.S.P.2023.000.31230. The GTL license on the code (https://github.com/CL-CHEN-Lab/MnM/blob/main/LICENSE) is an open-source license for academics, while Industrial/commercial use of the software is restricted and requires a separate agreement. This software was developed using Python and its associated libraries, in compliance with their respective licences. The R script to discover qualitative barcodes from single cells through the expectation-maximisation algorithm, the Python script to split subpopulation and replicate copy number files, the related code scRT files and scCNV matrices from the data can be found at the MnM GitHub depository. Kronos scRT under GNU General Public License (GPL-3.0 – 2007/06/29) can be found at https://github.com/CL-CHEN-Lab/Kronos_scRT and the modified Kronos version used here can be found at https://github.com/josephides/Kronos_scRT.

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

## Acknowledgements

Part of this manuscript was prepared using a limited access dataset obtained from BC Cancer and does not necessarily reflect the opinions or views of BC Cancer. The authors would like to acknowledge Tatiana Popova and Guillem Rigaill for stimulating discussions, Elisabetta Andermarcher for critical reading of the manuscript, Jean-Baptiste Guillaumin and Emeline Ravalli for the APP application, and Yoann Schumacher for establishing the programme license. C.L.C's team was supported by the ATIP-Avenir programme from the Centre national de la recherche scientifique (CNRS) and Plan Cancer from INSERM [ATIP/AVENIR: No 18CT014-00], the Agence Nationale pour la Recherche (ANR) [ReDeFINe – 19-CE12-0016-02, TELOCHROM – 19-CE12-0020-02, SMART – 21-CE12-0033-02, ReSPoND – 23-CE12-0020-02], the Institut National du Cancer (INCa) [PLBIO19-076] and the Impulscience programme of the Bettencourt Schueller Foundation. J.M.J. was supported by a PSL-Qlife fellowship [ANR-17-CONV-0005].

## Author contributions

C.L.C. conceived and planned the study. J.M.J. developed the programme and performed the bioinformatics analyses. C.L.C. supervised the development and bioinformatics analyses. J.M.J. and C.L.C. wrote the manuscript.

## Competing interests

The authors declare no competing interests.
