## [Transparent Peer Review file · Nature Communications]

Unravelling single-cell DNA replication timing dynamics using machine learning reveals heterogeneity in cancer progression

Corresponding Author: Dr Chun-Long Chen

Version 0:

Reviewer comments:

Reviewer #1

(Remarks to the Author)

This manuscript by Josephides and Chen describes a machine-learning tool, MnM, that efficiently establishes a means to determine single-cell replication states and detect subpopulations from cell lines and patient samples. Their machine-learning approach first utilizes KNN imputation to account for missing copy number values and then uses a feed forward neural network to classify/predict replication states and detect subpopulation using unsupervised clustering. The authors highlight higher accuracy of their model in determining DNA replication state when compared to FACS sorting. The observations of superior accuracy were made using data from three cell lines. Furthermore, the authors reveal genomic heterogeneity/subpopulation detection and determine single cell replication timing profiles for each subpopulation (as well as timing profile correlations between subpopulations from the same sample) in two cancer cell lines and one patient sample.

The overall topic of the manuscript and its future applications are promising, and the manuscript is well written. The authors have demonstrated the efficiency and accuracy of their machine-learning tool in detecting single-cell replication timing profiles within distinct subpopulations from DNA copy-numbers across heterogenous cancer cell lines. However, their validation in patient tumors is limited to a single patient sample. Finally, the creation of a scRT atlas holds the potential to further improve understanding of DNA replication timing and aneuploidy, and their contributions to cancer progression.

I just have a few questions/comments that I would like to be addressed:

1. In the introduction, the authors mention limitations in terms of assessing genomic heterogeneity from bulk whole genome sequencing studies. While there are limitations, there are numerous studies performed with innovative mathematical and computational approaches that can infer subclonal structure from bulk tumor sequencing data by estimating tumor purity and leveraging variant allele fractions of somatic variants.
2. The authors empirically validate their imputation method by simulating sparse single-cell copy number matrices – this is done by introducing random voids within a matrix of MCF7 cells. The authors may benefit from using an additional cell line harboring a large number of CNAs (with more sparsity/voids) to validate the accuracy and robustness of the KNN imputation predictions. Considering, the authors used deep learning to create the classifier, why not use deep learning (eg. generative adversarial imputation net)?
3. While FACS sorting has its limitations and may have high error rates, much of this can be attributed to technical variations within experimental execution. The authors should compare the accuracy of their model with other FACS datasets available with different/better accuracies as well as other cell cycle analysis methods (i.e. DNA content microscopy analysis) to bolster their claim of superior accuracy rates.
4. Can the authors speculate why choosing a 500kb bin for training the model resulted in lower accuracy than 100kb window?
5. Please explain how the numbers of UMAP coordinates were determined to identify clusters. Did the authors consider other combinations of dimensionality reduction and clustering approaches? I wonder whether the selection of only lower dimensions might have excluded rare subpopulations. Along these lines, did the authors consider clustering raw data directly instead of using UMAP coordinates?
6. What was the advantage of using a feedforward neural network for constructing the classifier? Did the authors try any other machine learning architectures that also provide information about feature relevance?
7. Did the authors evaluate how does the classifier perform with hyperploid cell lines (for example, some hepatocellular

carcinoma cell lines that exhibit chromosome numbers greater than 4 copies)?

8. My main concern is that the model training and validation were largely performed using data from cell lines, which are unaffected by normal/diploid subpopulations of cells that are typically found in patient tumors. The authors use only a single human tumor sample to demonstrate application of their method in patients. This is a very limited application of the method. The authors should include multiple tumor samples with a range of ploidy states/subpopulations to demonstrate robustness of the method.

9. The section describing the application of the scRT atlas could be expanded further with additional examples. Considering the inherent cost of scWGS, I am not sure whether determining cell type specificity alone is a particularly attractive application.

(Remarks on code availability)

The authors have provided code, example datasets and instructions for installation of required packages/resources to reproduce their work.

Reviewer #2

(Remarks to the Author)

Review report of NCOMMS-24-09776

In this paper, Josephides and Chen (abbreviated as JC in the sequel) proposed a new machine-learning-based tool for inferring single-cell DNA replication timing (scRT for short), and as a result, formed a “database” for scRT profiles across 0.1 million human cells. In view of the prevalence of single cell technologies and the need to understand the heterogeneous cancer etiology, there is no doubt that JC have made a timely contribution to the literature on the molecular biological mechanism of cancer development. The paper looks technically sound. That being said, I hope that the authors could consider to further improve the written quality of this paper. In particular, I provided some of my personal suggestions below, together with some other comments on the manuscript. I hope that my suggestions can be useful for the authors to further improve their paper.

Major comments and questions:

1. The biological significance of the result should be further delineated by highlighting several key results demonstrating that the new method MnM can provide additional and orthogonal biological knowledge compared to the case without using MnM.
2. In the second but last Result section (line 218), JC wrote “In some cases, we discovered large copy-number differences between the same cell lines that were obtained from different publications”. Could this be due to confounding batch effects?
3. To continue, in the same section, JC wrote “This was determined either by software failure or visual inspection of the replication patterns and manual elimination”. Are “visual inspection” and “manual elimination” justified by any means?
4. Again, JC wrote “we discovered that this cell line only shared between 11 and 13 common median chromosome copies between the two sample origins”. Is this conclusion supported by any other orthogonal evidence?
5. The titles of several subsections are not very well-written. The first result section: “Highly accurate data completeness for unsynchronised single-cell copy-numbers” looks a bit confusing. I recommend that “data completeness” be replaced by “data imputation” or “data completion”; The second result section: “Deep learning single-cell DNA replication state classifier with high accuracy” is recommended to be changed to “Deep learning classifies single-cell DNA replication states with high accuracy”.
6. Overall, the language of the paper should be further edited. As it currently stands, there are still numerous typos (see below) and sentences with broken English.
7. The last result section introducing MnM seems to contain very little extra information compared to what have been covered in the manuscript. I would personally recommend that this section be moved to either the Discussion section, or be moved to the very beginning of the results part.
8. Finally, several earlier works on DNA replication timing and cancer mutation based on bulk data are recommended to be referenced, including De and Michor, Nat. Biotech. 2011, and to a lesser degree, Liu, De and Michor, Nat. Comm. 2013; Fudenberg, Getz, Meyerson, Mirny, Nat. Biotech. 2011; Schuster-Bockler and Lehner, Nat. 2014.

Some minor typos:

1. Line 90: “sparse” should be “sparse”.
2. Line 114: “supplementary information” is probably better rephrased as “auxiliary information”.
3. Line 152: “second dimensionality reduction step in 10 dimensions” should probably be “second dimensionality reduction step down to 10 dimensions”.
4. Line 165: “previously publication” should be “previous publication”.
5. Line 173: “rather that” should be “rather than”.
6. Line 236: “single” is probably better rephrased as “stand-alone”.
7. Line 238: “Copy-number imputation, replication state classification and subpopulation detection enabled ...” is probably better rephrased as “Copy-number imputation, replication state classification and subpopulation detection, taken together, enabled ...”.
8. Line 308: “then (ii) through an unsupervised ...” should be “then (iii) an unsupervised ...”.
9. Line 226: “Knowing that this cell line is known to have variable karyotypes” should be better rephrased as “Knowing that this cell line has variable karyotype”.

(Remarks on code availability)

The code provides a user-friendly readme file as instructions for installation and implementation. Overall it is easy to use and may have potential to have positive impact on the field.

Version 1:

Reviewer comments:

Reviewer #1

(Remarks to the Author)

The authors have sufficiently addressed my queries. In particular, they have included additional data from multiple cell lines which improves the reliability of the results and the overall quality of the manuscript.

(Remarks on code availability)

I have no additional comments beyond the initial review.

Reviewer #2

(Remarks to the Author)

The authors have sufficiently addressed my comments. Thank you for the due diligence.

(Remarks on code availability)

I went through the github page quickly. One extra thing I would like the authors to add is the example code for replicating the results in the paper.

Reviewer #3

(Remarks to the Author)

(Remarks on code availability)

Reviewer #4

(Remarks to the Author)

(Remarks on code availability)

POINT BY POINT RESPONSE TO REVIEWERS' COMMENTS:

Overview:

We thank both reviewers for their suggestions and constructive comments that were very helpful for improving the manuscript. We are now submitting a revised manuscript. Based on the reviewers' suggestions, the revision clarifies and strengthens our conclusions, as well as reports additional observations and analysis results to address the reviewers' concerns. The major changes in the manuscript file are highlighted (in red), and a point-by-point response to each reviewer's comments (in blue) is included below.

REVIEWER COMMENTS

Reviewer #1 (Remarks to the Author): Expert in bioinformatics, AI/machine learning, cancer genomics, and single-cell omics

General comment: This manuscript by Josephides and Chen describes a machine-learning tool, MnM, that efficiently establishes a means to determine single-cell replication states and detect subpopulations from cell lines and patient samples. Their machine-learning approach first utilizes KNN imputation to account for missing copy number values and then uses a feed forward neural network to classify/predict replication states and detect subpopulation using unsupervised clustering. The authors highlight higher accuracy of their model in determining DNA replication state when compared to FACS sorting. The observations of superior accuracy were made using data from three cell lines. Furthermore, the authors reveal genomic heterogeneity/subpopulation detection and determine single cell replication timing profiles for each subpopulation (as well as timing profile correlations between subpopulations from the same sample) in two cancer cell lines and one patient sample.

The overall topic of the manuscript and its future applications are promising, and the manuscript is well written. The authors have demonstrated the efficiency and accuracy of their machine-learning tool in detecting single-cell replication timing profiles within distinct subpopulations from DNA copy-numbers across heterogenous cancer cell lines. However, their validation in patient tumors is limited to a single patient sample. Finally, the creation of a scRT atlas holds the potential to further improve understanding of DNA replication timing and aneuploidy, and their contributions to cancer progression.

Our response: We would like to thank the reviewer for the positive support of our study. We appreciate the reviewer's consideration of the potential impact and future applications of our work.

I just have a few questions/comments that I would like to be addressed:

Comment 1. In the introduction, the authors mention limitations in terms of assessing genomic heterogeneity from bulk whole genome sequencing studies. While there are limitations, there are numerous studies performed with innovative mathematical and computational approaches that can infer subclonal structure from bulk tumor sequencing data by estimating tumor purity and leveraging variant allele fractions of somatic variants.

Our response: We thank the reviewer for the suggestion, we modified the text to include this important information and the corresponding references in the Introduction section as follows: "Although efforts have been made to develop innovative mathematical and computational approaches to estimate tumour purity and leverage variant allele frequency in bulk sequencing data, the accurate detection of intra-

tumoral genomic heterogeneity^{20,21}, an important aspect of evolving tumour populations^{11,22}, is still challenging because bulk data do not provide single-cell resolution.” (Page 1-2).

Comment 2.1. The authors empirically validate their imputation method by simulating sparse single-cell copy number matrices – this is done by introducing random voids within a matrix of MCF7 cells. The authors may benefit from using an additional cell line harboring a large number of CNAs (with more sparsity/voids) to validate the accuracy and robustness of the KNN imputation predictions.

Our response: We thank the reviewer for the suggestion. In addition to the MCF-7 cell line (mean ploidy: 3.73), we repeated this part of the analysis using HeLa cells (mean ploidy: 2.87) and JEFF lymphocytes (mean ploidy: 1.94) for the evaluation of the KNN imputation method with various copy number alteration levels. We added these results in the Results section and the supplementary section (Supplementary Fig. 1b and c) as follows “We repeated the evaluation of the KNN imputation method using scWGS data of HeLa S3 cervical cancer cells (mean ploidy: 2.87, n=459) and JEFF B cells (mean ploidy: 1.94, n=952) (Supplementary Fig. S1b-c) and obtained similar invariance rates (99.28%-91.33% and 99.61%-95.43%, respectively). These random simulation results confirmed that our KNN imputation method can robustly produce accurate results using scWGS data obtained from normal and cancer cell types with various ploidy levels.” (Page 3).

Comment 2.2. Considering, the authors used deep learning to create the classifier, why not use deep learning (eg. generative adversarial imputation net)?

Our response: Although deep learning could be an interesting approach to detect missing values in the copy number matrix, due to the low rate of missing values in our data, which are presented in Supplementary Table 1 (median = 0.34%), we considered that a simplistic approach, such as KNN, would perform well enough without the drawback of a high computational cost. We added a sentence in the Discussion section to underline that, in addition to KNN, other methods such as Generative Adversarial Nets could also be well-performing approaches. “Considering that the data used in our current study present a low rate of missing copy number values (median = 0.34% in the present study), the KNN approach used here is robust enough for imputation without a high computational cost. For more sparse data, other deep learning approaches (such as Generative Adversarial Nets)^{48,49} could also be considered.” (Page 7).

Comment 3. While FACS sorting has its limitations and may have high error rates, much of this can be attributed to technical variations within experimental execution. The authors should compare the accuracy of their model with other FACS datasets available with different/better accuracies as well as other cell cycle analysis methods (i.e. DNA content microscopy analysis) to bolster their claim of superior accuracy rates.

Our response: We agree with the reviewer that technical variations can indeed lead to differences in error rates between FACS experiments. To address this limitation, we revised the Discussion section to recognize that our tool should be further tested as new datasets become available. “Our study underlines the FACS limitations for cell cycle phase detection (at least for the scWGS data with available FACS sorting data included in the current study), revealing error rates that ranged from 17.7% to 27.7%, in accordance with previous estimations³⁸. Although in some cases single cells are sorted into the different cell cycle phases before sequencing, and the latest version of cell sorting machines/methods can provide more accurate results, we believe that cell-sorted metadata should always be verified computationally to avoid erroneous conclusions.”...“Nonetheless, we acknowledge that the amount of data used here to compare our model is limited and as a result, it would be interesting to further test our model using new scWGS data as they become available.” (Page 7).

Comment 4. Can the authors speculate why choosing a 500kb bin for training the model resulted in lower accuracy than 100kb window?

Our response: The difference in accuracy with models trained with different bin sizes might be due to two possible factors, one biological and one technical. Biologically, smaller replication domains would not be identified with the 500 kb model. Technically, during the training of the sequential model, shuffling training data during each epoch and any other stochastic process embedded in the training process could induce small changes to the results. To better quantify the difference between models, we ran a proportion test and observed no significant difference (p-value 0.3904) between the accuracy rates of 98.54% and 98.14% for the two models with different bin sizes.

Comment 5. Please explain how the numbers of UMAP coordinates were determined to identify clusters. Did the authors consider other combinations of dimensionality reduction and clustering approaches? I wonder whether the selection of only lower dimensions might have excluded rare subpopulations. Along these lines, did the authors consider clustering raw data directly instead of using UMAP coordinates?

Our response: The number of UMAP coordinates was determined after evaluating different numbers of UMAP coordinates. The number of clusters did not increase when testing between 2 and 16 UMAP coordinates for the MCF-7, JEFF, HeLa, and T47D cell lines as well as the SA1135 TNBC tumour samples. Nonetheless, to address this point, we plan to allow the user to decide the number of coordinates in the next release of MnM. We did consider clustering the raw data directly; however, the runtime was extended, and for practical reasons, we did not evaluate this approach. In comparison to other dimensionality reduction techniques, “UMAP provides fastest run times, highest reproducibility and the most meaningful organization of cell clusters” as reported in Becht et al. 2018 (Nat Biotech). One of the advantages that led us to select DBSCAN was because it can discover arbitrarily shaped clusters, which can be the case of our CNV UMAPs, as illustrated in Figures 3 and 5. Additionally, unlike k-means and other clustering algorithms, DBSCAN does not require the number of clusters as input, thus enabling the automation of the clustering process. Furthermore, we determined that the combination of these algorithms was sufficient to detect subpopulations because the numbers of subpopulations identified were concordant with the numbers in the original publications (Gnan et al. 2022; Laks et al. 2019) and could also be verified by visually inspecting the genome-wide CNA landscapes in Supplementary Figure 3.

We agree with the reviewer that selecting lower dimensions might have excluded some rare subpopulations (if they exist); however, it is also likely that such rare subpopulations may not contain enough cells to extract meaningful replication timing information. Because the interest of our paper is on replication timing, we did not explore this, but users of MnM will be able to do this in the future.

Comment 6. What was the advantage of using a feedforward neural network for constructing the classifier? Did the authors try any other machine learning architectures that also provide information about feature relevance?

Our response: The advantage of the feedforward network is that it is one of the simplest to implement deep learning architectures. Deep learning has the advantage of automatically selecting the features (genomic regions) that are important for the model. Although not reported here, we also tested other machine learning methods, such as the Random Forest algorithm, but they did not provide reproducibility on all datasets, nor high accuracy. Overall, we were confident of the results of our trained model, and we were able to visually identify the copy number differences between cells in S and G1 phase, as illustrated in Supplementary Figure 3G. Nevertheless, we do not exclude that other algorithms

could provide even better models, and we hope that our findings will encourage such explorations in the future.

Comment 7. Did the authors evaluate how does the classifier perform with hyperploid cell lines (for example, some hepatocellular carcinoma cell lines that exhibit chromosome numbers greater than 4 copies)?

Our response: In the current study, we used all the available published high-throughput scCNV data containing S-phase cells that we could find. We used the MCF-7 cell line that is close to tetraploidy (mean ploidy: 3.73). As in our previous publication (Gnan et al. 2022), we confirmed the presence of subpopulations with different CNA patterns in MCF7 cells with other means, we were able to study this cell line with an extra layer of certainty. Unfortunately, at the current stage, we did not have any other hyperploid cell line with average chromosome numbers >4 copies to further test our methods. On the other hand, the HeLa cell line and the tumour samples that we analysed in the current study were also aneuploid, and importantly, the numbers of subpopulations obtained with our method matched the numbers reported in the original publication (Laks et al. 2019).

Comment 8. My main concern is that the model training and validation were largely performed using data from cell lines, which are unaffected by normal/diploid subpopulations of cells that are typically found in patient tumors. The authors use only a single human tumor sample to demonstrate application of their method in patients. This is a very limited application of the method. The authors should include multiple tumor samples with a range of ploidy states/subpopulations to demonstrate robustness of the method.

Our response: The reviewer is right that the model training and validation were largely performed using data from cell lines. However, as these are high-quality data validated in previous studies, they can be used as a positive control in the current study. Although the model training was performed using data from cell lines, for the validation, we mixed data from various cell lines with both normal/diploid cells (such as JEFF lymphocytes) and aneuploid cells (such as HeLa and MCF7 cells) to mimic what happens inside patient tumours. As indicated in the response to Comment 7, the model did accurately separate data from different cell types and different subpopulations of MCF7 cells. In addition, the accuracy of our model was directly validated with a breast cancer patient sample. Furthermore, to prevent the model from overfitting our cell line data, we implemented an early stopping mechanism during the training process. Based on the reviewer's suggestion, to further extend and explore the application of our methods, we performed additional scRT trajectory analyses from data of various cell lines and tumour samples. We added the new results in the Results section as follows with new figures (Fig. 7 and Supplementary Fig. 4):

“To better examine the similarities and differences of the various RT profiles of the scRT atlas, we generated RT trajectories using potential of heat-diffusion for affinity-based trajectory embedding (PHATE)⁴⁷ to visualise the local and also the global structure of the RT landscapes. Initially, we generated the RT trajectories in the well-characterised cell lines of this study (Fig. 7a, c) and found distinct RT dynamics for different cell types, consistent with previous findings³⁷. Notably, the trajectories of the lymphocyte lines (i.e. JEFF, GM12891, G12892) and HCT-116 cell subpopulations reflected the closeness of correlations observed in the scRT atlas (Fig. 6), whereas MCF-7 cell subpopulations showed slight divergences, as previously reported³⁷. We then extended this analysis to the breast cancer samples and PDXs of the atlas (Fig. 7b, d). We observed a general conservation of RT profiles by tumour; however, we also detected RT profile divergences during PDX passaging (e.g. TNBC PDX SA609), reflecting RT profile evolution during cancer progression. Notably, the trajectories uncovered subtle shifts in RT patterns, suggesting that RT profiling can capture genomic changes during tumour progression.

To assess whether there was a significant difference in RT trajectories among subpopulations, we collected the scRT trajectory coordinates from Figure 7b, grouped by subpopulation and used them for

a permutation test (see Methods), a non-parametric method that does not assume specific data distributions and allows the randomisation of group (i.e. cell-type/subpopulation) labels. In agreement with our previous report³⁷, the trajectories of the two MCF-7 cell subpopulations were significantly different (p-value = 0.004). We generated pairwise p-values for each (sub)population (Supplementary Fig. S4) and demonstrated that the RT trajectories of some tumour samples (e.g. TNBC PDX SA501 and PDX SA532, ER-positive breast cancer) did not change through PDX passaging, while in other samples (e.g. TNBC PDX SA609) they constantly changed. Furthermore, the breast cancer sample SA1135 showed an RT trajectory change between subpopulations. Altogether, these results show that RT subpopulation extraction can be used to study RT dynamics during cancer progression in vitro (i.e. cell lines) and in vivo or ex vivo (i.e. tumour/PDX samples).” (Page 6).

Comment 9. The section describing the application of the scRT atlas could be expanded further with additional examples. Considering the inherent cost of scWGS, I am not sure whether determining cell type specificity alone is a particularly attractive application.

Our response: We thank the reviewer for the suggestion. As indicated in the response to Comment 8, we expanded the description of our atlas by adding the scRT trajectory analyses and comparisons of various cell lines and tumour samples. In addition, to better emphasize the importance of obtaining directly the scRT data from tumour samples and/or subpopulations for future studies, we added the following sentences in the Discussion section: “Many pioneering studies revealed the important links between the DNA RT programme and cancer mutation rates, signatures and structural variants^{9,44,62–64}. However, due to the lack of RT data directly measured in tumour samples and despite a general association observed between the RT programme and genetic alterations in cancer, few studies have been able to analyse how the RT programme plasticity and changes interact with the mutational landscape in cancer and contribute to cancer development. In a recent study⁴⁴, Dietzen and colleagues focused on RT alterations by comparing RT data from lung and breast cancer cell lines and matched normal cell lines. They showed that RT is altered relatively early during cancer development/progression and that this plays an important role in shaping the cancer mutational landscape. Interestingly, they found that regions changing from late to early replication contain genes that are overexpressed in cancer and present a preponderance of APOBEC3-mediated mutation clusters associated with driver mutations. These results suggest a direct role of RT in tumour development/progression.” ... “Until now, only RT data from cancer cell lines were available and previous studies neglected the intra- and inter-tumour heterogeneity. Here, we provide a new tool that allowed generating unique data on subpopulation-resolved RT profiles from tumour samples. These results show that this tool can facilitate the exploration of the intra-tumour RT programme heterogeneity and its relationship with genomic alterations in individual patient tumours.” (Page 7-8).

Reviewer #1 (Remarks on code availability):

The authors have provided code, example datasets and instructions for installation of required packages/resources to reproduce their work.

Our response: We would like to thank the reviewer for the positive evaluation on the quality and availability of our code.

Reviewer #2 (Remarks to the Author): Expert in computational cancer genomics, DNA replication timing estimation, single-cell omics, and AI/machine learning

General comment: In this paper, Josephides and Chen (abbreviated as JC in the sequel) proposed a new machine-learning-based tool for inferring single-cell DNA replication timing (scRT for short), and as a result, formed a “database” for scRT profiles across 0.1 million human cells. In view of the prevalence of single cell technologies and the need to understand the heterogeneous cancer etiology, there is no doubt that JC have made a timely contribution to the literature on the molecular biological mechanism of cancer development. The paper looks technically sound. That being said, I hope that the authors could consider to further improve the written quality of this paper. In particular, I provided some of my personal suggestions below, together with some other comments on the manuscript. I hope that my suggestions can be useful for the authors to further improve their paper.

Our response: We would like to thank the reviewer for the positive support of our study. We appreciate the reviewer’s consideration of the timely contribution and impact of our work.

Major comments and questions:

Comment 1. The biological significance of the result should be further delineated by highlighting several key results demonstrating that the new method MnM can provide additional and orthogonal biological knowledge compared to the case without using MnM.

Our response: We thank the reviewer for the suggestion. To highlight the biological significance and the interest of our scRT atlas, we expanded the description of our atlas by adding the scRT trajectory analyses and comparisons of various cell lines and tumour samples. We added the new results in the Results section as follows with new Figures (Fig. 7 and Supplementary Fig. 4):

“To better examine the similarities and differences of the various RT profiles of the scRT atlas, we generated RT trajectories using potential of heat-diffusion for affinity-based trajectory embedding (PHATE)⁴⁷ to visualise the local and also the global structure of the RT landscapes. Initially, we generated the RT trajectories in the well-characterised cell lines of this study (Fig. 7a, c) and found distinct RT dynamics for different cell types, consistent with previous findings³⁷. Notably, the trajectories of the lymphocyte lines (i.e. JEFF, GM12891, G12892) and HCT-116 cell subpopulations reflected the closeness of correlations observed in the scRT atlas (Fig. 6), whereas MCF-7 cell subpopulations showed slight divergences, as previously reported³⁷. We then extended this analysis to the breast cancer samples and PDXs of the atlas (Fig. 7b, d). We observed a general conservation of RT profiles by tumour; however, we also detected RT profile divergences during PDX passaging (e.g. TNBC PDX SA609), reflecting RT profile evolution during cancer progression. Notably, the trajectories uncovered subtle shifts in RT patterns, suggesting that RT profiling can capture genomic changes during tumour progression.

To assess whether there was a significant difference in RT trajectories among subpopulations, we collected the scRT trajectory coordinates from Figure 7b, grouped by subpopulation and used them for a permutation test (see Methods), a non-parametric method that does not assume specific data distributions and allows the randomisation of group (i.e. cell-type/subpopulation) labels. In agreement with our previous report³⁷, the trajectories of the two MCF-7 cell subpopulations were significantly different (p-value = 0.004). We generated pairwise p-values for each (sub)population (Supplementary Fig. S4) and demonstrated that the RT trajectories of some tumour samples (e.g. TNBC PDX SA501 and PDX SA532, ER-positive breast cancer) did not change through PDX passaging, while in other samples (e.g. TNBC PDX SA609) they constantly changed. Furthermore, the breast cancer sample SA1135 showed an RT trajectory change between subpopulations. Altogether, these results show that RT subpopulation extraction can be used to study RT dynamics during cancer progression in vitro (i.e. cell lines) and in vivo or ex vivo (i.e. tumour/PDX samples).” (Page 6).

In addition, to better emphasize the importance of obtaining directly the scRT data from tumour samples and/or subpopulations for future studies, we added the following sentences in the Discussion section: “Many pioneering studies revealed the important links between the DNA RT programme and cancer mutation rates, signatures and structural variants^{9,44,62–64}. However, due to the lack of RT data directly measured in tumour samples and despite a general association observed between the RT programme and genetic alterations in cancer, few studies have been able to analyse how the RT programme plasticity and changes interact with the mutational landscape in cancer and contribute to cancer development. In a recent study⁴⁴, Dietzen and colleagues focused on RT alterations by comparing RT data from lung and breast cancer cell lines and matched normal cell lines. They showed that RT is altered relatively early during cancer development/progression and that this plays an important role in shaping the cancer mutational landscape. Interestingly, they found that regions changing from late to early replication contain genes that are overexpressed in cancer and present a preponderance of APOBEC3-mediated mutation clusters associated with driver mutations. These results suggest a direct role of RT in tumour development/progression.” ... “Until now, only RT data from cancer cell lines were available and previous studies neglected the intra- and inter-tumour heterogeneity. Here, we provide a new tool that allowed generating unique data on subpopulation-resolved RT profiles from tumour samples. These results show that this tool can facilitate the exploration of the intra-tumour RT programme heterogeneity and its relationship with genomic alterations in individual patient tumours.” (Page 7-8).

Comment 2. In the second but last Result section (line 218), JC wrote “In some cases, we discovered large copy-number differences between the same cell lines that were obtained from different publications”. Could this be due to confounding batch effects?

Our response: We thank the reviewer for pointing out this possibility. To better discuss this point, we added additional information and description in the Results section as follows: “It is unlikely that these differences were the result of the different scWGS techniques used for data generation or of confounding batch effects because the HCT-116 cells from two different publications^{32,38} had the same karyotype. Importantly, the observed results from HeLa and MCF-7 cells are in line with previous studies reporting karyotypic heterogeneity among and within different HeLa⁴⁵ and MCF-7 strains⁴⁶.” (Page 5-6).

Comment 3. To continue, in the same section, JC wrote “This was determined either by software failure or visual inspection of the replication patterns and manual elimination”. Are “visual inspection” and “manual elimination” justified by any means?

Our response: We clarified this sentence in the revised manuscript to explain that we needed cells to cover mid-S phase or both early and late S phase in the ploidy/copy-number variability plots, such as those in Figure 2. This information has been added in the Results section as follows: “We computed the pseudo-bulk RT profiles when we had enough cells to reconstruct a representative S-phase landscape. This was determined either by software failure or visual inspection of the ploidy/copy-number variability plots (i.e. Fig. 2) to make sure that cells were either in mid-S phase or in early and late S phase. When this was not the case, we did not consider the (sub)populations for RT profile generation.” (Page 6).

Comment 4. Again, JC wrote “we discovered that this cell line only shared between 11 and 13 common median chromosome copies between the two sample origins”. Is this conclusion supported by any other orthogonal evidence?

Our response: In our previous publication (Gnan 2022), we reported the existence of two MCF-7 subpopulations issued from the same experiment. We did perform Fluorescence In Situ Hybridization (FISH) to verify the existence of these two subpopulations and to quantify their proportions. This additional experiment allowed us to confirm that the copy-number profiles that we detected *in silico* could also be seen *in vitro*. We added this information in the Results section as follows: “Importantly, the observed results from HeLa and MCF-7 cells are in line with previous studies reporting karyotypic heterogeneity among and within different HeLa⁴⁵ and MCF-7 strains⁴⁶. In addition, the presence of these two MCF-7 cell subpopulations was confirmed experimentally by FISH³⁷.” (Page 5-6).

Comment 5. The titles of several subsections are not very well-written. The first result section: “Highly accurate data completeness for unsynchronised single-cell copy-numbers” looks a bit confusing. I recommend that “data completeness” be replaced by “data imputation” or “data completion”; The second result section: “Deep learning single-cell DNA replication state classifier with high accuracy” is recommended to be changed to “Deep learning classifies single-cell DNA replication states with high accuracy”.

Our response: We thank the reviewer for these suggestions. We agree with the reviewer and modified the titles accordingly.

Comment 6. Overall, the language of the paper should be further edited. As it currently stands, there are still numerous typos (see below) and sentences with broken English.

Our response: The revised manuscript has been proof-read by native English speakers.

Comment 7. The last result section introducing MnM seems to contain very little extra information compared to what have been covered in the manuscript. I would personally recommend that this section be moved to either the Discussion section, or be moved to the very beginning of the results part.

Our response: We thank the reviewer for the suggestion, and we moved this section to the very beginning of the Results section, as suggested.

Comment 8. Finally, several earlier works on DNA replication timing and cancer mutation based on bulk data are recommended to be referenced, including De and Michor, Nat. Biotech. 2011, and to a lesser degree, Liu, De and Michor, Nat. Comm. 2013; Fudenberg, Getz, Meyerson, Mirny, Nat. Biotech. 2011; Schuster-Bockler and Lehner, Nat. 2014.

Our response: We thank the reviewer for the references, which now have been included in the revised manuscript.

Comment 9. Some minor typos:

1. Line 90: “sparce” should be “sparse”.
2. Line 114: “supplementary information” is probably better rephrased as “auxiliary information”.
3. Line 152: “second dimensionality reduction step in 10 dimensions” should probably be “second dimensionality reduction step down to 10 dimensions”.
4. Line 165: “previously publication” should be “previous publication”.

5. Line 173: “rather that” should be “rather than”.
6. Line 236: “single” is probably better rephrased as “stand-alone”.
7. Line 238: “Copy-number imputation, replication state classification and subpopulation detection enabled ...” is probably better rephrased as “Copy-number imputation, replication state classification and subpopulation detection, taken together, enabled ...”.
8. Line 308: “then (ii) through an unsupervised ...” should be “then (iii) an unsupervised ...”.
9. Line 226: “Knowing that this cell line is known to have variable karyotypes” should be better rephrased as “Knowing that this cell line has variable karyotype”.

Our response: We thank the reviewer, and we corrected the corresponding typos as suggested by the reviewer. In addition, the revised manuscript has been proof-read by native English speakers.

Reviewer #2 (Remarks on code availability):

The code provides a user-friendly readme file as instructions for installation and implementation. Overall it is easy to use and may have potential to have positive impact on the field.

Our response: We would like to thank the reviewer for the positive evaluation of our code availability, and we appreciate the reviewer’s consideration of the potential impact and future applications of our tool in the field.